# Game bird carcasses are less persistent than raptor carcasses, but can predict raptor persistence dynamics

Eric Hallingstad[1]☯*, Daniel Riser-Espinoza[2]☯, Samantha Brown[3]‡, Paul Rabie[4]‡, Jeanette Haddock[1]‡, Karl Kosciuch[1]‡

**1** Western EcoSystems Technology, Inc., Cheyenne, Wyoming, United States of America, **2** Western EcoSystems Technology, Inc., Fort Collins, Colorado, United States of America, **3** Western EcoSystems Technology, Inc., Corvallis, Oregon, United States of America, **4** Western EcoSystems Technology, Inc., Laramie, Wyoming, United States of America

☯ These authors contributed equally to this work.
‡ SB, PR, JH and KK also contributed equally to this work.
* ehallingstad@west-inc.com

**Data Availability Statement:** All relevant data are within the paper and its Supporting Information files.

## Abstract

Researchers conduct post-construction fatality monitoring (PCFM) to determine a wind energy facility's direct impacts on wildlife. Results of PCFM can be used to evaluate compliance with permitted take, potentially triggering adaptive management measures or offsetting mitigation; reducing uncertainty in fatality rates benefits wind companies, wildlife agencies, and other stakeholders. As part of PCFM, investigators conduct carcass persistence trials to account for imperfect detection during carcass surveys. In most PCFM studies, pen-raised game birds and other non-raptor surrogates have been used to estimate persistence of all large birds, including raptors. However, there is a growing body of evidence showing carcass persistence varies by bird type; raptor fatality estimates based on game bird carcass persistence may therefore be biased high. We conducted raptor and game bird carcass persistence field trials for 1 year at 6 wind energy facilities. Raptor carcass persistence varied by habitat and season, whereas the best-supported game bird model only included habitat. Raptor persistence probabilities were higher than corresponding game bird persistence probabilities for 13 of the 16 habitat and season combinations. Analysis of a curated large bird persistence meta-dataset showed that raptor carcass persistence varied by season, habitat, and region. The probability of persisting through a 30-day search interval ranged from 0.44 to 0.99 for raptors and from 0.16 to 0.79 for game birds. Raptor persistence was significantly higher than game bird persistence for 95% of the sampled strata. We used these carcass persistence estimates to develop linear mixed-effects models that predict raptor persistence probabilities based on estimated game bird persistence probabilities. Our scaling model provides an important statistical method to address gaps in raptor persistence data at sites in a broad range of landscape contexts in the continental United States and should be used to inform fatality estimation when site-specific raptor persistence data are limited or absent.

**Funding:** Funding for this research was provided to the authors (WEST, Inc.) by the Renewable Energy Wildlife Research Fund (REWRF; https://rewi.org/2022/03/24/research-fund-expands-to-solar-becomes-renewable-energy-wildlife-research-fund/). The funders had no role in data collection and analysis, or preparation of the manuscript. REWRF members reviewed the manuscript, approved the use of their existing data, and participated in the decision to publish.

**Competing interests:** I (EH) have read the journal's policy and the authors of this manuscript have the following competing interests: funding for the research was provided by the Renewable Energy Wildlife Research Fund. REWRF members reviewed the manuscript, approved the use of their existing data, and participated in the decision to publish. All authors work for Western EcoSystems Technology, Inc., an environmental consulting firm. There are no patents, products in development, or marketed products to declare. The above disclosures do not alter the authors' adherence to all the PLOS ONE policies on sharing data and materials, as detailed online in the guide for authors.

## Introduction

The risk of birds colliding with wind turbines, especially sensitive or protected species such as bald eagle (*Haliaeetus leucocephalus*) or golden eagle (*Aquila chrysaetos*), is a fundamental challenge the wind industry faces when developing and operating facilities. Wind companies are either compelled by permit conditions (e.g., eagle incidental take permit) or voluntarily complete post-construction fatality monitoring (PCFM) to estimate a facility's direct impacts on birds [1]. Imperfect detection of carcasses necessitates the use of statistical methods to calculate fatality rates (e.g., fatalities/turbine/year), which provide results that could trigger adaptive management measures or require mitigation. Thus, reducing uncertainty in fatality rates will benefit wind companies, wildlife agencies, and other stakeholders.

Calculating fatality rates requires that several detection bias parameters are measured or estimated [1–4] to understand the overall probability that a carcass is detected at a wind facility. A carcass might be missed by searchers because they failed to detect it, or the carcass was scavenged (i.e., removed) prior to the search, the carcass fell outside of the search area, or the turbine was not searched. The product of these bias parameters is used to adjust the number of carcasses detected. If the bias parameters are low (i.e., low detection, short persistence, small search area), large scaling of the number of carcasses found to the number of fatalities estimated can occur, resulting in uncertainty around the actual number of fatalities at a wind facility.

One factor that influences detection probability is carcass persistence; carcass persistence probability can be less than 1.0 due to scavenging or other processes (e.g., human activities) that remove carcasses from the landscape prior to carcass searchers having the opportunity to detect those carcasses. Carcass persistence can be measured experimentally in the field with carcass persistence trials (CPT), where intact, fresh carcasses are placed at a facility and monitored to determine how long the carcasses remain discoverable in the field. In most PCFM studies (including studies required by eagle incidental take permits), pen-raised game birds and other non-raptor surrogates have been used as trials to estimate persistence probabilities of all large birds, including raptors [3, 5]. However, game birds consistently have shorter persistence times than large raptor carcasses, which would result in a lower probability of persistence and a higher fatality estimate if other parameters are unchanged [6–9]. Therefore, raptor fatality estimates based on carcass persistence data collected from game bird species may be biased high [9].

Overestimation of fatality rates can trigger compensatory mitigation or the implementation of costly avoidance and minimization measures to "reduce take" to remain within permitted levels. Eagle carcasses and parts are protected by federal law and are prioritized for Native American religious purposes, so obtaining authorization to use eagle carcasses for site-specific bias trials is unlikely. Fresh, intact non-eagle raptor carcasses can also be difficult to obtain, and the permitting hurdles to procure these carcasses often result in game bird carcasses being used for site-specific CPT, despite the known limitations. Existing data on raptor carcass persistence are limited in geographic and habitat representation, and few studies have measured game bird and raptor carcass persistence simultaneously. Moreover, most previous CPT studies of raptors (or other species) have focused on data collected at a single site, or a small number of sites, over a limited period [3, 7–12], and there had not been a thorough investigation of the potential trends in persistence data from raptors or between bird types (but see Wilson et al. [13]). However, if a relationship between game bird and raptor persistence is predictable and can be quantified, site-specific game bird carcass persistence could be adjusted to provide a more accurate prediction of raptor persistence for a wind facility.

To fill some of the regional knowledge gaps in raptor carcass persistence data and analyze the relationships between raptor and game bird carcass persistence, we conducted a carcass

persistence study across 4 United States Fish and Wildlife Service (USFWS) Regions (collectively, Regions) and aggregated those data with a larger dataset of raptor and game bird CPT from across the United States (U.S.). The primary objectives of our study are to: 1) estimate and compare persistence times and probabilities for large raptor and game bird carcasses placed concurrently at 6 wind facilities in a variety of landscape contexts; 2) evaluate patterns in large raptor and game bird persistence among Regions, habitats, and seasons in a meta-analysis using data curated from a broad range of studies; and 3) develop a model to determine the predictive relationship between large raptor and game bird persistence probabilities using CPT data from studies across the U.S.

## Study areas

Four Regions were selected for CPT to capture a spatial distribution across the U.S., and to target geographic areas and/or habitat types with little available information on raptor carcass persistence (Regions 2, 4, 5, and 6; Fig 1). Region 2, also referred to as the Southwest Region, is characterized by semiarid and temperate climates, and consists primarily of grasslands,

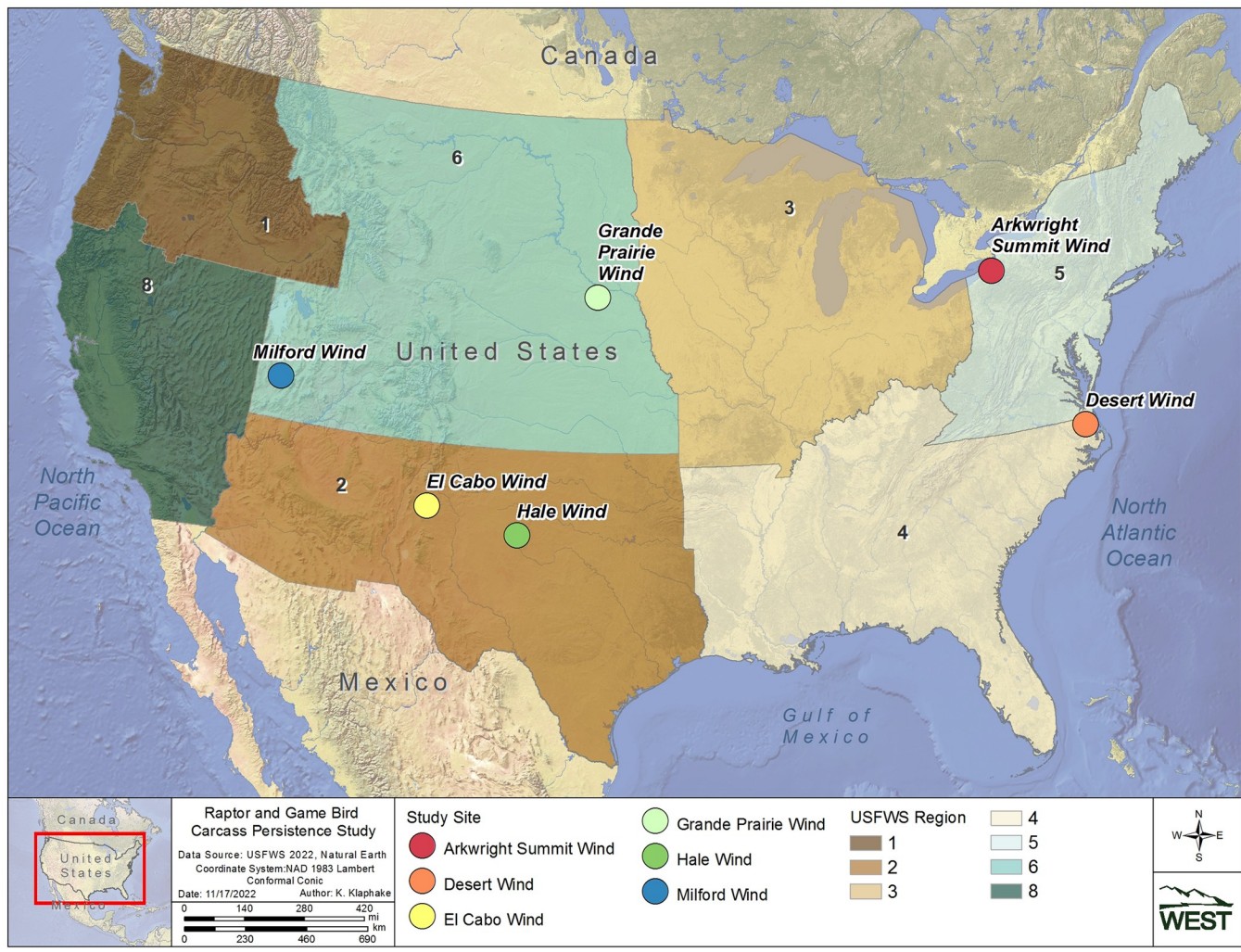

**Fig 1. Study site locations for the carcass persistence study conducted in USFWS regions 2, 4, 5, and 6 from June 2020 to August 2021.**

**Table 1. Study site descriptions for the carcass persistence study conducted from June 2020 to August 2021.**

| Study Site | USFWS Region | Megawatts | Level IV Ecoregion | Primary Land Use | Predominant Habitat | Source |
|---|---|---|---|---|---|---|
| Hale Wind Farm (Hale) | 2 | 478 | Llano Estacado | Crop production (e.g., cotton [*Gossypium spp.*] and corn [*Zea mays*]); cattle (*Bos taurus*) grazing | Cropland | [14, 16] |
| El Cabo Wind Farm (El Cabo) | 2 | 298 | Central New Mexico Plains | Cattle grazing | Grassland | [17, 18] |
| Desert Wind Farm (Desert Wind) | 4 | 208 | Chesapeake-Pamlico Lowlands and Tidal Marshes | Crop production (e.g., soybean [*Glycine max*], corn, cotton) | Cropland | [19–21] |
| Arkwright Summit Wind Farm (Arkwright) | 5 | 79 | Low Lime Drift Plain | Dairy cow production | Forest | [22, 23] |
| Grande Prairie Wind Farm (Grande Prairie) | 6 | 400 | Holt Tablelands | Crop production (e.g., corn, soybean) and grassland prairie | Grassland | [24, 25] |
| Milford Wind Corridor (Milford) | 6 | 306 | Sagebrush Basins and Slopes | Cattle grazing | Shrub/scrub | [26–30] |

deserts, and diverse forests in areas of higher elevations in Arizona and New Mexico [14]. Region 4, or the Southeast Region, is characterized by a subtropical humid climate and is dominated by deciduous forests and wetlands [15]. Region 5, also known as the North Atlantic-Appalachian Region, is characterized by a continental humid climate and is primarily dominated by deciduous forests with some open wetland areas [15]. Region 6, commonly referred to as the Mountain Prairie Region, is characterized by a semiarid climate and is dominated by temperate grasslands and savannas, with diverse deciduous and conifer forests at higher elevations.

We conducted the field component of our study at 6 different wind energy facilities (Study Sites), across for different Regions (see Table 1, Figs 1 and 2). We assigned habitat associations (e.g., cropland, grassland, forest, and shrub/scrub) for each Study Site based on the predominant land cover present at each Study Site and its surrounding area, as we anticipated this would likely influence the composition of local scavenger communities.

## Methods

To meet our first objective, we conducted CPT at the 6 Study Sites (Table 2) to estimate and directly compare persistence metrics for large raptor and game bird carcasses placed concurrently in a variety of landscape contexts. To meet our second and third objectives regarding a broader evaluation of the patterns in raptor and game bird persistence, we conducted 2 meta-analyses using a curated meta-dataset containing information from suitable PCFM studies covering a broad range of habitats and geographic regions within the continental U.S.

### Field trial methods

Our study was divided into 4 seasons: summer (June 15 to September 14), fall (September 15 to November 14), winter (November 15 to March 14), and spring (March 15 to June 15). In each season, 10 large raptor and 10 game bird carcasses were placed at each Study Site (Table 2) in land cover types representative of the turbine locations and monitored over 56 days. At Arkwright, Grande Prairie, Hale, and El Cabo, each carcass was checked in-person. At Milford, carcasses were monitored continuously by game cameras (i.e., rugged, weather-proof, motion-triggered cameras) throughout the 56-day trials. At Desert Wind, in-person checks were conducted during the fall, winter, and spring CPT, while game cameras were used during the summer CPT to accommodate limited biologist availability.

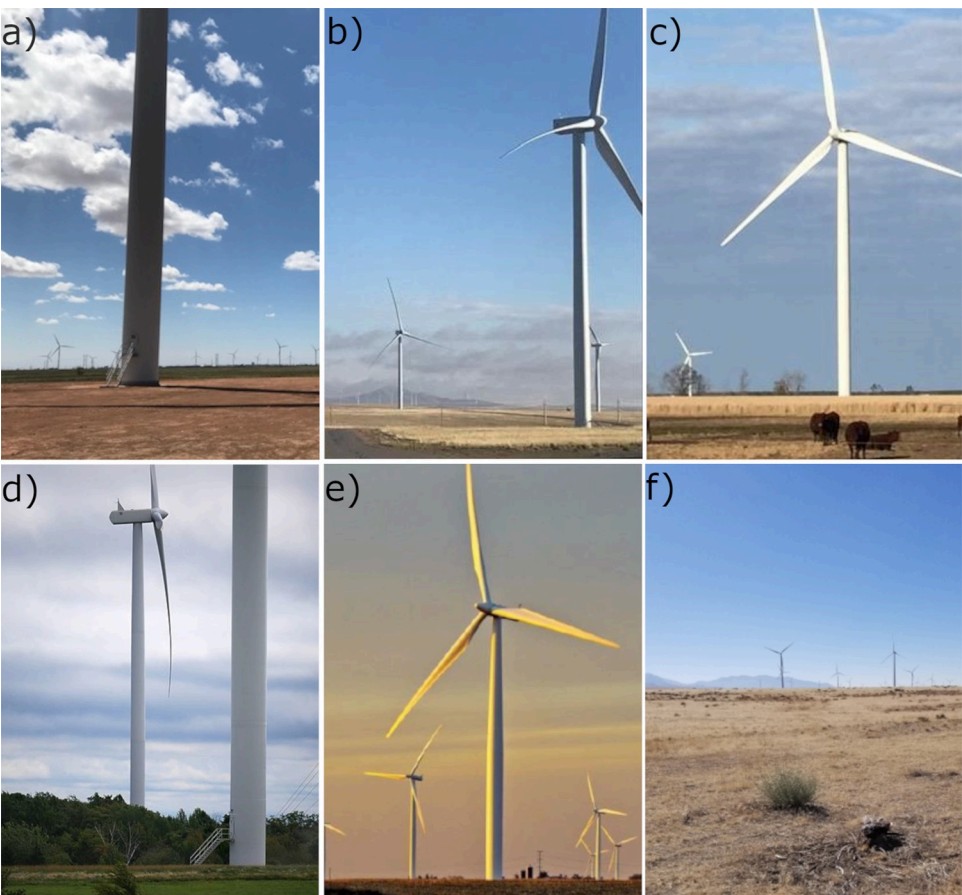

**Fig 2. The study sites where large raptor and game bird carcasses were placed concurrently for carcass persistence trials.** a) Hale, b) El Cabo, c) Desert Wind, d) Arkwright, e) Grande Prairie, and f) Milford.

At Arkwright, Grande Prairie, Hale, El Cabo, and Milford, our study began in June 2020 and ended May 2021, whereas CPT at Desert Wind started in September 2020 and ended August 2021 due to permit delays. A Federal Migratory Bird Special Purpose—Utility permit is required for the use of raptor carcasses in CPTs at a wind facility. Delays were due to regional USFWS differences in permit conditions and language, and the need for an amendment after the permit was issued.

Trial carcasses for game birds consisted of hen or immature male (i.e., drab-colored and similar in size to a hen) ring-necked pheasants (*Phasianus colchicus*) and hen mallards (*Anas*

**Table 2. Number of carcass persistence trial carcasses concurrently placed in each USFWS region, study site, and habitat type during June 2020 –August 2021 field trials for our study.**

| Region | Study Sites | Habitat | Game Bird | Raptor |
|--------|-------------|---------|-----------|--------|
| **Region 2** | El Cabo | grassland | 40 | 40 |
| | Hale | cropland | 40 | 40 |
| **Region 4** | Desert Wind | cropland | 40 | 40 |
| **Region 5** | Arkwright | forest | 40 | 40 |
| **Region 6** | Milford | grassland | 40 | 39 |
| | Grande Prairie | shrub/scrub | 40 | 40 |
| **Total** | | | **240** | **239** |

*platyrhynchos*). Trial carcasses for raptors included adult and fully grown juvenile red-tailed hawk (*Buteo jamaicensis*), great horned owl (*Bubo virginianus*), barn owl (*Tyto alba*), Cooper's hawk (*Accipiter cooperii*), ferruginous hawk (*Buteo regalis*), prairie falcon (*Falco mexicanus*), Swainson's hawk (*Buteo swainsoni*), rough-legged hawk (*Buteo lagopus*), peregrine falcon (*Falco peregrinus*), osprey (*Pandion haliaetus*), and turkey vulture (*Cathartes aura*). Carcasses were not weighed prior to deployment, but we set a minimum raptor size threshold of an approximately 30-cm wing chord and a 300-g mass (e.g., a male short-eared owl [*Asio flammeus*]) in order to balance our objective of estimating generic "large" raptor persistence with sample size requirements. Comparatively, our game bird species typically have had average masses exceeding 900 g. All trial carcasses were intact, with no evidence of decomposition, infestation, or disease; no euthanizing agents were used on any trial carcasses. All carcasses were obtained from either wildlife rehabilitators or airports implementing lethal (but non-toxic) control measures with proper permit authorizations. Carcasses were frozen and distributed to the Study Sites seasonally. Carcass availability limited our ability to equally deploy species both spatially and temporally. Carcasses were thawed and placed at random coordinates within the leased boundary of each designated Study Site at a distance of 200 m or greater from turbine locations to avoid potential risks to aerial scavengers and minimize confusion between study carcasses and any other fatalities near turbines. When necessary, trial carcasses were placed within 200 m of turbines for consistency with ongoing PCFM protocols (e.g., summer and fall CPT at 4 Study Sites); however, all protocols accommodated drop locations in representative habitats. All carcasses were marked discreetly with dark electrical tape for recognition by searchers and other personnel. Carcasses were dropped from waist height or higher and allowed to land in a random posture. At the end of the CPT period, all remaining evidence of trial carcasses was removed from the field.

**In-person checks.** At Arkwright, Desert Wind, Grande Prairie, Hale, and El Cabo, we monitored the trial carcasses over a 56-day trial period, with checks scheduled every 7 days for 56 days following placement; however, schedules varied slightly depending on weather and other unforeseen circumstances. Universal Transverse Mercator (UTM) location, date and time placed, species, age, and sex were recorded for each trial carcass. During each carcass check, the time and date of check was recorded, as well as the condition of the carcass (i.e., intact, partial, feather spot, absent). Intact meant there was no evidence of scavenging, partial meant that there was a partial carcass remaining, feather spot meant that there were at least 10 feathers and/or 2 flight feathers remaining, and absent meant that the carcass was either not present or that there were fewer than 10 contour feathers and 2 flight feathers remaining (consistent with protocols described by Strickland et al. [2]).

If a carcass was not in its original placement location during a check, the area was searched systematically out to 30 m from the point of placement to determine if any evidence or parts of the carcass were still present. If the carcass had been moved due to scavenging activity, new UTM locations and photographs of the carcasses were recorded. Trial carcasses were considered absent and categorized as unavailable for detection (i.e., removed from the landscape) as soon as fewer than 2 flight feathers and 10 contour feathers remained.

**Camera checks.** Due to remoteness of the Study Site, game cameras were implemented at Milford to monitor trial carcasses. Game cameras have been shown to be a viable alternative to in-person checks, providing a cost-effective approach with more precise carcass removal timing without biasing fatality estimates compared to in-person checks [31]. Cameras were inconspicuously placed at or near ground level approximately 3 m from each carcass, with little to no visual obstruction between the camera and carcass. Cameras were secured using stakes to prevent movement during weather events or due to curious animal interference (e.g., livestock). Cameras were checked every 2 weeks when possible (depending on schedule

availability and weather), or once a month at minimum, to download pictures and replace batteries as needed during the 56-day trial period. After pictures were downloaded, a field technician reviewed the photographs and filled out datasheets for each designated check day. When a removal event occurred between designated check days, photographs were reviewed to determine the smallest interval between last known time the carcass was present and first known time the carcass had been removed, or the actual removal date was recorded. At Desert Wind, game cameras were used during the summer CPT following the methods described above because of limited availability of local field staff.

## Analysis methods

We conducted a separate analysis to meet each of the 3 primary objectives of our study, and used different techniques appropriate for data exploration (Objective 1 and Objective 2) and prediction (Objective 3; see Tredennick et al. [32]). In the first analysis (hereafter, Field Trial Analyses), we analyzed raptor and game bird persistence data from the concurrent CPT at the 6 Study Sites during 2020–2021 to develop models of persistence. To meet Objective 2, we used data we had curated from a meta-dataset of available studies and the same modeling techniques used in the first analysis to explore broad patterns of raptor and game bird persistence (hereafter, Raptor and Game Bird Persistence Meta-analysis). In the third analysis, we used Raptor and Game Bird Persistence Meta-analysis probabilities from studies that included both bird types, and fit models to determine a predictive relationship between game bird and raptor persistence probabilities (hereafter, Scaling Game Bird Persistence). All analyses were conducted in R [33]. When comparing estimates, we considered a significant difference at the α = 0.10 significance level if the 90% confidence intervals (CI) did not overlap.

**Field trial analyses.** To compare persistence times and probabilities for large raptor and game bird carcasses placed concurrently in a variety of landscape contexts, data collected during our CPT were analyzed using interval-censored survival regression in the GenEst package [34–37], as is typical for PCFM studies. The interval-censored survival regression models, denoted $S(t)$, were used to estimate median persistence time and average probability of persistence $(\widehat{r})$ for time interval $t$, where:

$$\widehat{r} = \frac{\int_0^t S(t)dt}{t}. \tag{1}$$

There are numerous metrics related to carcass persistence that can be generated from interval-censored survival regression models. We focused on the median estimated persistence time (in days) and average probability of persistence metrics because median persistence time provides an intuitive measure of how long carcasses are expected to remain detectable on the landscape, while average probability of persistence contributes directly to the development of detection probabilities and fatality estimates in the context of PCFM search effort. For time interval $t$, we used 30, 60, and 90 days to approximate search intervals we expect will be most commonly used in eagle fatality monitoring studies.

Candidate models were fit using exponential, log-logistic, lognormal, and Weibull survival distributions to characterize a broad range of persistence behaviors. Separate models were fit for raptors and game birds to allow for potentially different distributions for each bird type. Habitat (cropland, forest, grassland, and shrub/scrub) and season (spring, summer, fall, winter) were considered as covariates of interest and we fit models that included 1 or both covariates on the location and scale parameters used to define the distributions above; see [34–36] for details about the location and scale parameterizations used for the candidate survival distributions. Covariates on the location parameter quantify the influence of those covariates on

where the distribution is centered (i.e. the median estimated persistence time), whereas covariates on the scale parameter quantify the influence of those covariates on the spread or shape of the distribution, and both parameters affect the average probability of persistence. We fit all possible combinations of covariates with interaction terms, but discarded models that did not have the main effects involved in higher-order interactions. We used sample-size corrected Akaike's Information Criterion (AICc) to rank models [38]. We selected the most parsimonious model within 2 AICc points of the top model (based on AICc rank). We used the selected model to produce estimates of median persistence time (in days), and average probability of persistence for 30-, 60-, and 90-day search intervals, and 90% CIs on each metric. The CIs were calculated using parametric bootstrapping.

**Meta-dataset curation.** Available raptor and game bird persistence data were compiled by running a query of the WEST project database for post-construction monitoring projects from 2010 through early 2021. For any projects with large bird CPT (including raptors and/or game birds), persistence data were included in the analyses for Objectives 2 and 3, provided we had permission to use the data in this analysis. If permissions were granted, we added the applicable projects to the dataset (S1 Table).

We reviewed the complete list of species in the initial dataset and filtered to include only "large" raptors (a minimum of approximately 30-cm wing chord and 300-g mass) or game birds. The retained raptor species included 12 species that were used in our field trials: red-tailed hawk, barn owl, osprey, turkey vulture, great horned owl, peregrine falcon, Cooper's hawk, Swainson's hawk, ferruginous hawk, prairie falcon, rough-legged hawk, and short-eared owl. An additional 7 large raptor species were included in the larger dataset: red-shouldered hawk (*Buteo lineatus*), barred owl (*Strix varia*), northern harrier (*Circus hudsonius*), northern goshawk (*Accipiter gentilis*), black vulture (*Coragyps atratus*), broad-winged hawk (*Buteo platypterus*), and snowy owl (*Bubo scandiacus*). The game bird species included mallard and ring-necked pheasant, both of which were used in our field trials, as well as ring-necked duck (*Aythya collaris*) and unidentified duck.

All raw data from the data curation effort (in addition to the data collected during our CPT) were compiled for analyses (see S1 Table). Trials were aggregated by study (the trial placement effort during a period up to a year, at a single facility), predominant habitat at the facility, Region, and season. We used the season assignments provided in the studies associated with each CPT, when available; when season assignments were absent, we assigned seasons based on the season dates used in the CPT's study (see above). Each study was also given an acronym of the form: Region-state abbreviation-habitat abbreviation-alpha numeric order (e.g., R3-IA-c-1 is the first study alphabetically in Region 3, in Iowa, with cropland as the predominant habitat).

Compiling the meta-dataset from wind facilities across the U.S. generated a large sample size for modeling. In general, game bird CPT had a trial length of at least 14 days, and raptor CPT had a trial length of at least 30 days. Typical CPT check schedules occurred on the first 4 days of a CPT, followed by days 7, 10, 14, 21, and 28; CPT extending longer were typically checked on days 35, 42, 49, and 56, followed by increasingly longer intervals between checks. We acknowledge CPT duration, time intervals between carcass checks, numbers of carcasses deployed per CPT, season dates, and other field methods varied throughout the dataset; however, the analytical methods we used (described above and below) are robust to the variation in the dataset [34]. Furthermore, a recent summary of persistence data no significant relationship between either the number of carcasses in the trial or trial duration and estimated carcass persistence [13].

**Raptor and game bird persistence meta-analysis.** To evaluate patterns in large raptor and game bird persistence among Regions, habitats, and seasons, we used the full meta-dataset

(including data collected during our field trials) to fit interval-censored survival regression using the GenEst package [34–37] for raptors and game birds. As in our Field Trial Analyses, we conducted model selection with exponential, log-logistic, lognormal, and Weibull survival distributions. With the larger dataset, potential covariates (on the location and/or scale parameter of the survival distribution) included season, Region, and habitat. We fit all combinations of covariates with interaction terms, but discarded any models without the main effects involved in higher-order interactions. We used sample-size corrected AICc to rank the models, selecting the most parsimonious model from among those within 2 AICc points of the top model. The selected models for raptors and game birds were used to estimate median persistence time and average probability of persistence for typical search intervals used in eagle fatality monitoring studies (30-, 60-, and 90-day intervals, with 30-day intervals assumed to be the most commonly used). Because we did not have data for all combinations of season, Region, and habitat for each CPT type, we only considered model output for the combinations of strata for which we did have data. We used parametric bootstrapping to develop 90% CIs for each metric.

**Scaling game bird persistence.** Our third objective was to develop a predictive model of raptor persistence probability based on game bird persistence probability. We filtered the curated dataset to only those sites with both raptor and game bird CPT data in order to pair estimates of raptor persistence with estimates of game bird persistence at the site level. Because average probability of persistence is a function of the model generated from persistence data on a site/study level, we needed a multi-step approach to get from interval-censored game bird persistence data to a scaled raptor average probability of persistence. We used a 2-stage modeling approach to: 1) generate game bird and raptor average probability of persistence estimates via interval-censored survival regression at the study level, and 2) model the relationship between raptor and game bird average probability of persistence estimates from the study-specific models (Fig 3; data sourced from [39–41]).

In Stage 1, we summarized the number of CPT by type (large raptor or game bird) from the filtered meta-dataset. To make the best use of the available data when there were fewer than 8 carcasses used in a study, we aggregated a low-sample size study with other studies (with both raptor and game bird trials) until a sample size of at least 8 carcasses was reached. When aggregation was necessary, studies were first aggregated by site (e.g., 2 studies at the same site in different years were combined), followed by aggregating with other sites in the same state (and thus Region) and habitat type. No analysis groups were created that included sites from multiple states; however, a single site included turbines located in two states. Hereafter, we refer to all studies and aggregated collections of studies to scale game bird persistence as "analysis groups". See S1 Table to identify trials by analysis groups.

After determining the analysis groups of raptor and game bird data, we fit separate interval-censored survival regression models [34–37] to the game bird and raptor data in each analysis group, consistent with the manner in which these data are typically analyzed in the context of PCFM. Candidate models for raptor and game bird persistence included exponential, log-logistic, lognormal, and Weibull survival distributions, which capture a wide range of persistence dynamics and are typical considerations in PCFM studies [35]. Some analysis groups contained data from a single season, while other contained data from multiple seasons at varying sample sizes We included season as a potential covariate (on the location and/or scale parameter) if there were at least 8 carcasses in each season represented in the analysis group. We used sample-size corrected AICc to rank models. We selected the most parsimonious model within 2 AICc points of the top model (based on AICc rank).

From each fitted model, we generated average probability of persistence for 14-, 30-, 60-, and 90-day intervals to capture a range of plausible search intervals and persistence

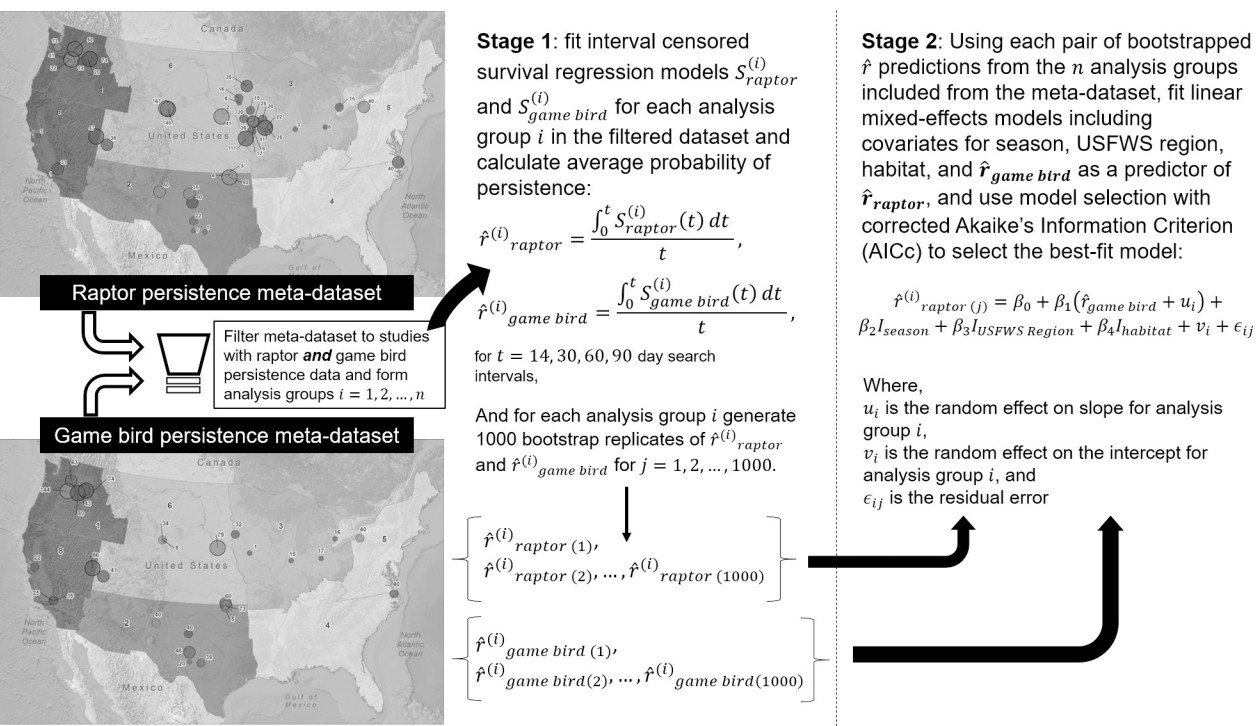

**Fig 3. The 2-stage analysis process to develop a model for scaling persistence probabilities ($\hat{r}$) from game birds to raptors using a meta-dataset curated from studies in the United States.**

probabilities in a monitoring context. To account for the uncertainty in estimated average probability of persistence from the first stage of the modeling process, we generated 1,000 sets of model parameters for each model using parametric bootstrapping in the GenEst package [36] and recalculated the persistence probabilities for each.

In Stage 2, we fit linear mixed effects models (LMEM [42]) implemented with the lme4 package [43] to predict raptor average probability of persistence as a function of game bird average probability of persistence. We used model selection with AICc to determine the best-fit model. Model selection included previously analyzed covariates (season, Region, habitat) as fixed effects to account for any systematic influence those variables might have on raptor and/ or game bird average probability of persistence. We did not include any interaction terms in our potential models due to limited sample sizes represented by combinations of Regions and habitats; thus, we were forced to assume interactions were negligible in the context of predicting raptor average probability of persistence based on game bird average probability of persistence. Another assumption of our model was that year was not an important predictor of raptor persistence in the context of game bird persistence measured in the same year, as we aggregated data by site from studies between 2010 and 2021. Analysis group was included as a random effect on both the intercept and slope to account for estimation uncertainty in the analysis group-specific average probability of persistence estimates from the first stage of the analysis. A logit transform was used on the response variable (raptor average probability of persistence) to assure the LMEM could return sensible predictions for any combination of predictors. Model selection was done by choosing the most parsimonious model within 2 AICc points of the top model.

After determining the best-supported model from the 2-stage approach above, we performed model validation by random, stratified cross-validation with 10 cross-validation sets.

For each cross-validation set, we randomly sampled a subset of the analysis groups in each combination of Region and habitat to use as a "training" set, and left the remaining analysis groups as an out-of-sample (OOS) set. When there was only 1 analysis group in a combination of Region and habitat, we always included that analysis group in the training set to avoid issues refitting the selected LMEM to the training set. When there was more than 1 analysis group in a combination of Region and habitat, the training set included between 1 and 10 analysis groups depending on the number of analysis groups in a combination of Region and habitat. The training set always contained 22, or 52%, of the analysis groups. We calculated 4 metrics comparing predicted raptor average probability of persistence to the raptor average probability of persistence for each study in the OOS set. Root mean-square error (RMSE) was a standardized measure of prediction error, Pearson's correlation coefficient was a measure of agreement between scaling model predictions and actual average probability of persistence, proportion of predictions exceeding the OOS estimates as a measure the direction of errors, and average absolute error (average of the absolute value of the difference between predicted and actual raptor average probability of persistence) was a raw measure of prediction error.

# Results

## Field trial results

To meet Objective 1, 479 trial carcasses were placed across the 6 Study Sites, of which 240 were game birds (10 per season per site) and 239 were raptors (10 per season per site; S2 Table). In 2020, carcasses were placed on June 15 for summer trials, September 15 for fall trials, and November 15 for winter trials; for spring trials, carcasses were placed March 15, 2021. To complete a full year of CPT at the Desert Wind site, summer trial carcasses were placed on June 15, 2021, as no CPT were conducted at Desert Wind during summer 2020.

**Raptor persistence.** Model selection for raptor persistence indicated carcass persistence varied by habitat and season (Delta [Δ] AICc 1.67; S3 Table). Median raptor persistence time was longest in grassland habitat during all seasons, with estimated median persistence times ranging from 74.3 days (90% CI 44.5–128) in spring to 184.9 days (90% CI 103.3–333.5) during summer (Fig 4, S4 Table). Conversely, median raptor carcass persistence time was shortest in forest habitat and ranged from 10.7 days in winter and spring (90% CI 6.8–17.4 and 6.1–17.8, respectively) to 26.6 days (90% CI 16.2–42.7) during summer (Fig 4, S4 Table). In the other habitats, median raptor persistence time estimates exceeded 47 days in all seasons. Median raptor persistence time was least variable (evaluated through confidence interval width) in forest habitat and most variable in grassland and shrub/scrub habitats (Fig 4, S4 Table).

Raptor persistence was longest during the summer season within each habitat type, with median persistence times exceeding 100 days in 3 of the 4 habitat types (S4 Table). Excluding forest habitat, median raptor persistence times were consistent, ranging from 47.6 days (90% CI 29.2–77.8) during spring in cropland habitat to 88.5 days (90% CI 51.4–154.4) during fall in grassland habitat (Fig 4, S4 Table). Variability in median persistence time was highest in the summer season, particularly in the grassland and shrub/scrub habitat sites (Fig 4, S4 Table).

Patterns in raptor average persistence probabilities were similar to those found for median persistence times. In all habitat types except forest habitat, the probability of a raptor persisting through a 30-day search interval was 0.77 or higher during all seasons (Fig 5, S4 Table). Within forest habitat, the probability of a raptor persisting through a 30-day search interval ranged from 0.46 (90% CI 0.35–0.58) during spring to 0.67 (90% CI 0.56–0.76) during summer (Fig 5, S4 Table). For a 60-day search interval, the probability of a raptor persisting was 0.64 or higher during all seasons in all habit types but forest (Fig 6, S4 Table). For a 90-day search interval,

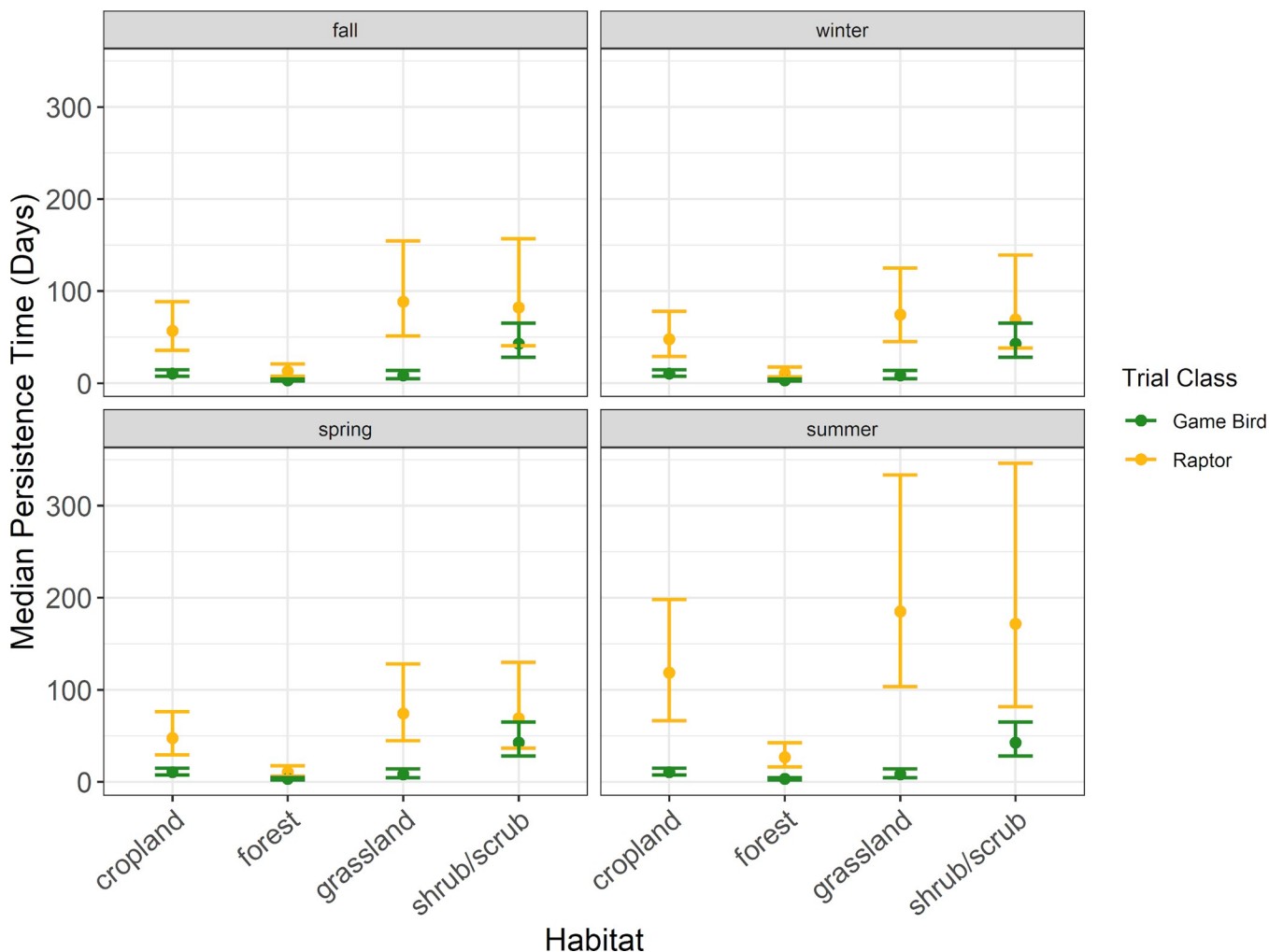

**Fig 4. The median persistence times of raptor and game bird carcasses among habitat types during the carcass persistence study conducted June 2020 – August 2021.**

the probability of a raptor persisting was 0.55 or higher during all seasons in all habitat types but forest (Fig 7, S4 Table).

**Game bird persistence.** The best supported model for game birds indicated carcass persistence varied among habitat types (Δ AICc 0; S5 Table); season was not included in the only model within 2 AICc points of the top model. Median game bird carcass persistence was longest in shrub/scrub habitat at 42.7 days (90% CI 28.0–65.2), followed by cropland, grassland, and forest habitats at 10.5 days (90% CI 7.5–14.8), 8.2 days (90 CI 4.8–14.0), and 3.2 days (90% CI 2.2–4.5), respectively (Fig 4, S6 Table).

Patterns in game bird average probability of persistence were similar to those found for median persistence rates. The highest probability of persistence to 30 days occurred in shrub/scrub habitat (0.78 [90% CI 0.70–0.85]), followed by cropland (0.48 [90% CI 0.41–0.55]), grassland (0.45 [90% CI 0.37–0.53]), and forest habitats (0.19 [90% CI 0.14–0.26]; Fig 4, S6 Table). The probability of a game bird carcass persisting to 60 days ranged from 0.11 (90% CI 0.07–0.15) in forest habitat to 0.64 (90% CI 0.53–0.73) in shrub/scrub habitat (S6 Table). For a 90-day search interval, the probability of a game bird persisting ranged from 0.07 (90% CI 0.05–0.10) in forest habitat to 0.54 (90% CI 0.42–0.64) in shrub/scrub habitat (S6 Table).

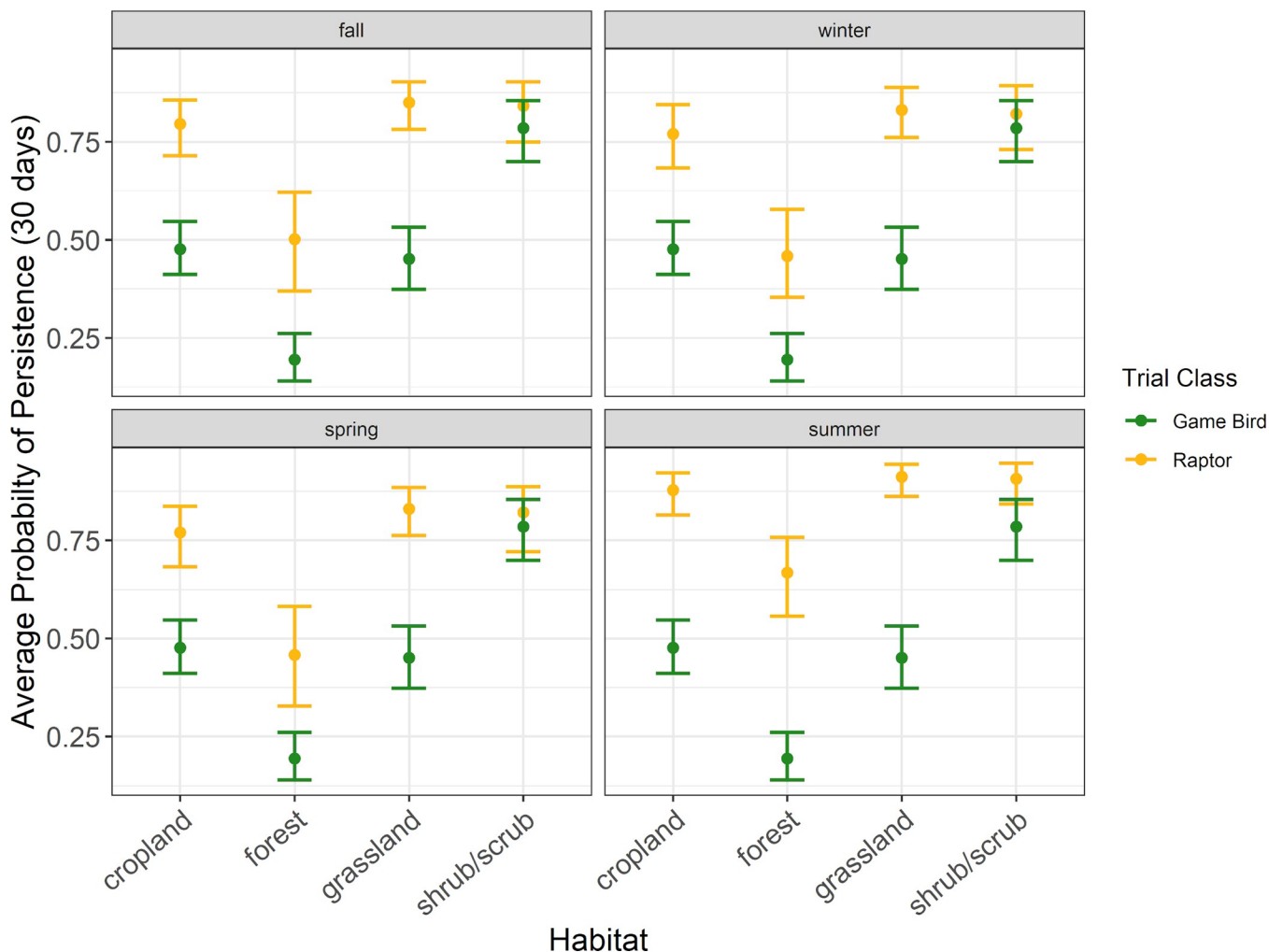

**Fig 5. The average probability a raptor or game bird carcass would persist for 30 days in 30 days in 4 habitat types during the carcass persistence study conducted June 2020 –August 2021.**

**Comparing field trial results for raptor and game bird persistence.** Within habitat types, we found that all point estimates for raptor average probability of persistence were higher than the corresponding point estimates for game bird average probability of persistence. The 90% CIs for raptor and game bird point estimates for 13 of the 16 habitat and season combinations in which we placed concurrent CPT did not overlap, suggesting significantly longer persistence times for raptors under most scenarios we tested in our field trials (Figs 4–7, S4 and S6 Tables). The 90% CIs overlapped between the bird types for median persistence and 30-day average probability of persistence within 1 habitat type (shrub/scrub) during 3 seasons (spring, winter, and fall; Figs 4–7, S4 and S6 Tables). Figures illustrating raptor and game bird average probability of persistence estimates for 30-, 60-, and 90-day intervals are provided (Figs 5–7), as these search intervals are currently the most common intervals considered in eagle fatality monitoring studies.

## Meta-dataset curation

Raptor and Game Bird Persistence Meta-analysis data were compiled from 100 studies at 45 wind facilities within 8 Regions (Figs 8 and 9; S1 Table). Trial data included 3,371 carcasses,

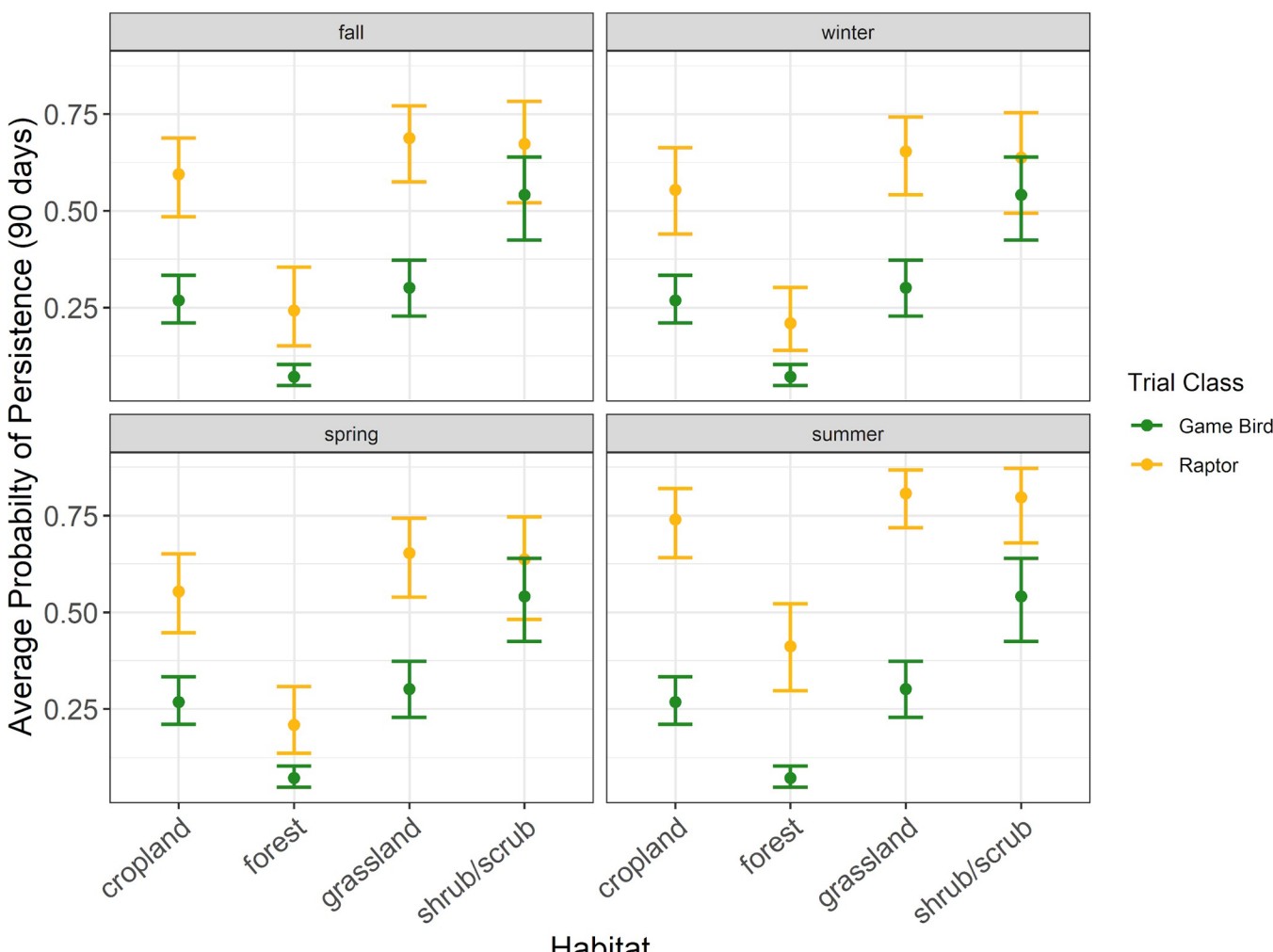

**Fig 6. The average probability a raptor or game bird carcass would persist for 60 days in 4 habitat types during the carcass persistence study conducted June 2020 –August 2021.**

1,624 raptors and 1,747 game birds, with carcasses placed during all months of the year. Raptors and game birds were represented across 7 Regions (1, 2, 3, 4, 5, 6, and 8), 4 habitats (cropland, forest, grassland, and shrub/scrub), and 4 seasons (fall, winter, spring, and summer; S7 and S8 Tables). Given the opportunistic nature of the meta-dataset curation, sample sizes were not balanced across all possible strata combinations. However, within every 2-variable combination of strata (i.e., Region and habitat, Region and season, and habitat and season), there were at least 20 carcasses for raptors and game birds, with 1 exception (13 raptor carcasses in Region 1, forest habitat; S9 and S10 Tables). The representation of seasons among carcasses was highest for winter (36.5%), followed by spring (23.8%), summer (21.3%), and fall (18.5%). Among studies, season assignments were relatively consistent and concentrated in 3-month intervals, with 94% of fall trials beginning in August–October, 87% of winter trials beginning December–February, 96% of spring trials beginning March–May, and 80% of summer trials beginning June–August. To meet Objective 2, the full meta-dataset provided information to estimate persistence metrics in at least 1 season for 14 of 32 combinations of Region and habitat (Fig 10; S1 Table). Of the 45 facilities, 23 contributed to raptor persistence data and 22

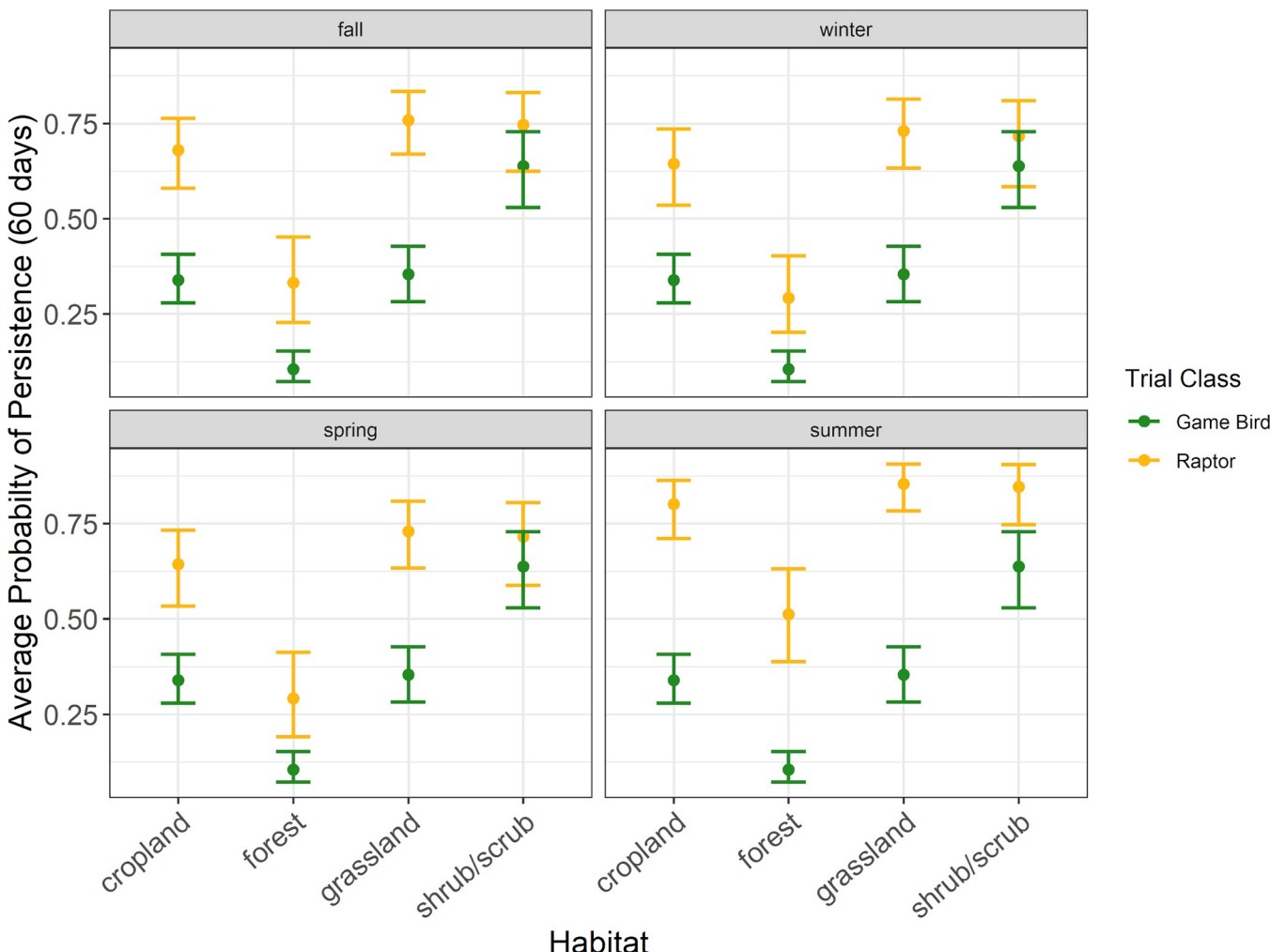

**Fig 7. The average probability a raptor or game bird carcass would persist for 90 days in 4 habitat types during the carcass persistence study conducted June 2020 –August 2021.**

contributed game bird carcass persistence data, totaling 2,018 carcasses (721 raptors and 1297 game birds) for use in the Objective 3 analysis to determine a scaling model between game bird and raptor persistence probabilities.

## Raptor and game bird persistence meta-analysis

**Raptor persistence meta-analysis.** Model selection for raptor persistence utilizing the full meta-dataset indicated raptor carcass persistence varied systematically by season, habitat, Region, and a season by Region interaction term on the location parameter (Δ AICc 0; S7 Table). Season and Region were covariates associated with the scale parameter in the top model (S7 Table). Season and/or Region also appeared as a covariate on both the location and scale parameters in all of the models within 10 AICc of the lowest AICc model, which further support the importance of temporal and spatial correlates of persistence (S7 Table). The median of all median raptor carcass persistence estimates was 63.4 days (S9 Table). Raptor persistence also showed considerable variability and was longest in shrub/scrub habitat during spring in Region 6 (median persistence time of 603.8 days; 90% CI 345.8–982.7; S9 Table).

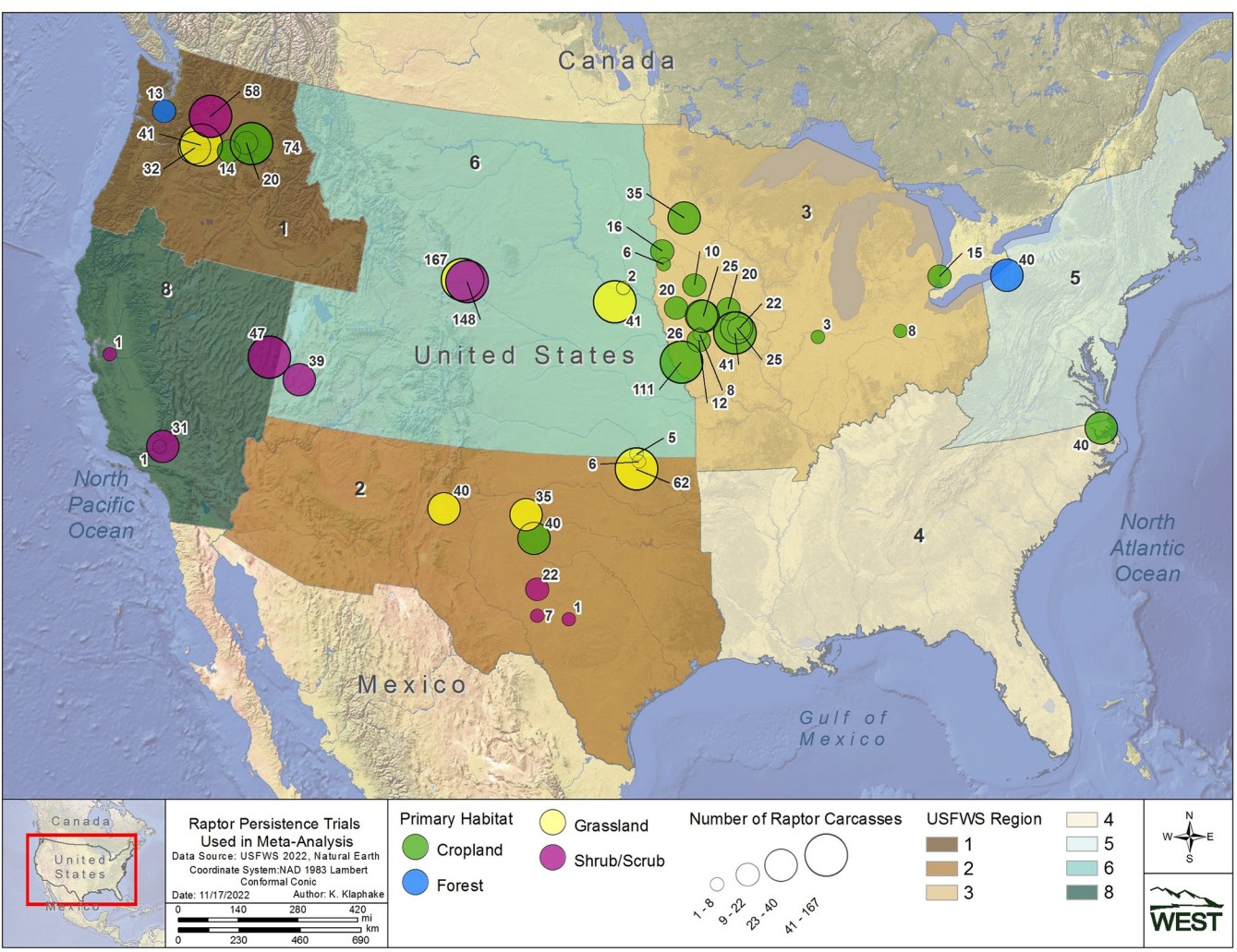

**Fig 8. Locations of raptor carcass persistence trials included in the meta-analyses.**

Conversely, median raptor carcass persistence was shortest in forest habitat within Region 5, at 11.3 days during both the fall (90% CI 6.7–19.3) and spring (90% CI 7.0-18.9; S9 Table).

Patterns of raptor average probability of persistence were generally consistent with patterns in median persistence time. The probability of a raptor persisting through a 30-day search interval was longest in shrub/scrub habitat in Region 8 during spring and summer, with median probabilities of persistence of 0.99 (90% CIs of 0.98–1.00 and 0.96–1.00, respectively; S9 Table) and shortest in forest habitat in Region 5 during spring, with a median probability of persistence of 0.44 (90% CI 0.30–0.62; Fig 10, S9 Table). The probability of a raptor carcass persisting through a search interval exceeded 0.50 in all but 2 of 49 strata combinations for a 30-day search interval, all but 6 strata combinations for a 60-day search interval, and all but 12 strata combinations for a 90-day search interval (S9 Table).

**Game bird persistence meta-analysis.** Model selection for game bird persistence utilizing the full meta-dataset indicated game bird carcass persistence times varied by season, habitat, Region, and a habitat by Region interaction term on the location parameter (Δ AICc 0; S8 Table). Region was the only covariate on the scale parameter of the top model and, in contrast to raptors, season was not a covariate on the scale parameter of any model (S8 Table). Game

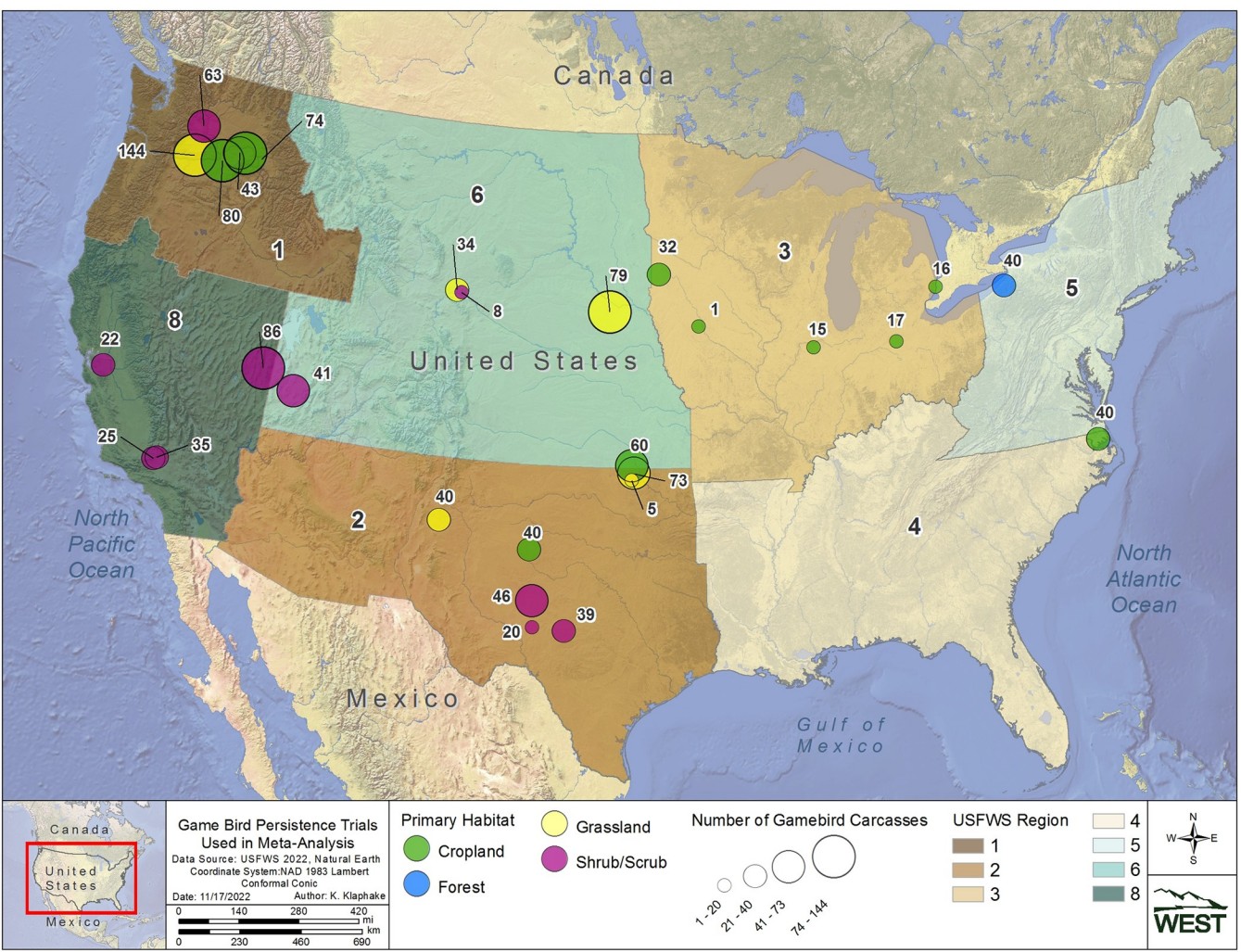

**Fig 9. Locations of game bird carcass persistence trials included in the meta-analyses.**

bird persistence was longest in shrub/scrub habitat during winter in Region 6, with a median persistence time of 48.2 days (90% CI 31.3–74.3; S10 Table). Conversely, median game bird carcass persistence was shortest in grassland habitat in Region 2 in the fall, at 2.5 days (90% CI 1.9–3.3; S10 Table).

The probability of a game bird carcass persisting through 30-, 60- and 90-day search intervals was consistent for the strata with the highest relative persistence and more variable for strata with the lowest relative persistence. The probability of a game bird persisting through a 30-day search interval ranged from 0.16 (90% CI 0.11–0.22) in forest habitat in Region 5 during fall to 0.79 (90% CI 0.71–0.85) in shrub/scrub habitat in Region 6 during winter (Fig 10, S10 Table). The probability of a game bird carcass persisting through a 30-day search interval was below 0.50 in all but 15 of 52 strata combinations for a 30-day search interval, all but 4 strata combinations for a 60-day search interval, and all but 3 strata combinations for a 90-day search interval (S10 Table).

**Patterns in raptor and game bird persistence.** Point estimates for raptor average probability of persistence were higher than game bird persistence point estimates in 60 out of 61 strata combinations (season by Region, by habitat type) for which we had meta-data (Fig 10,

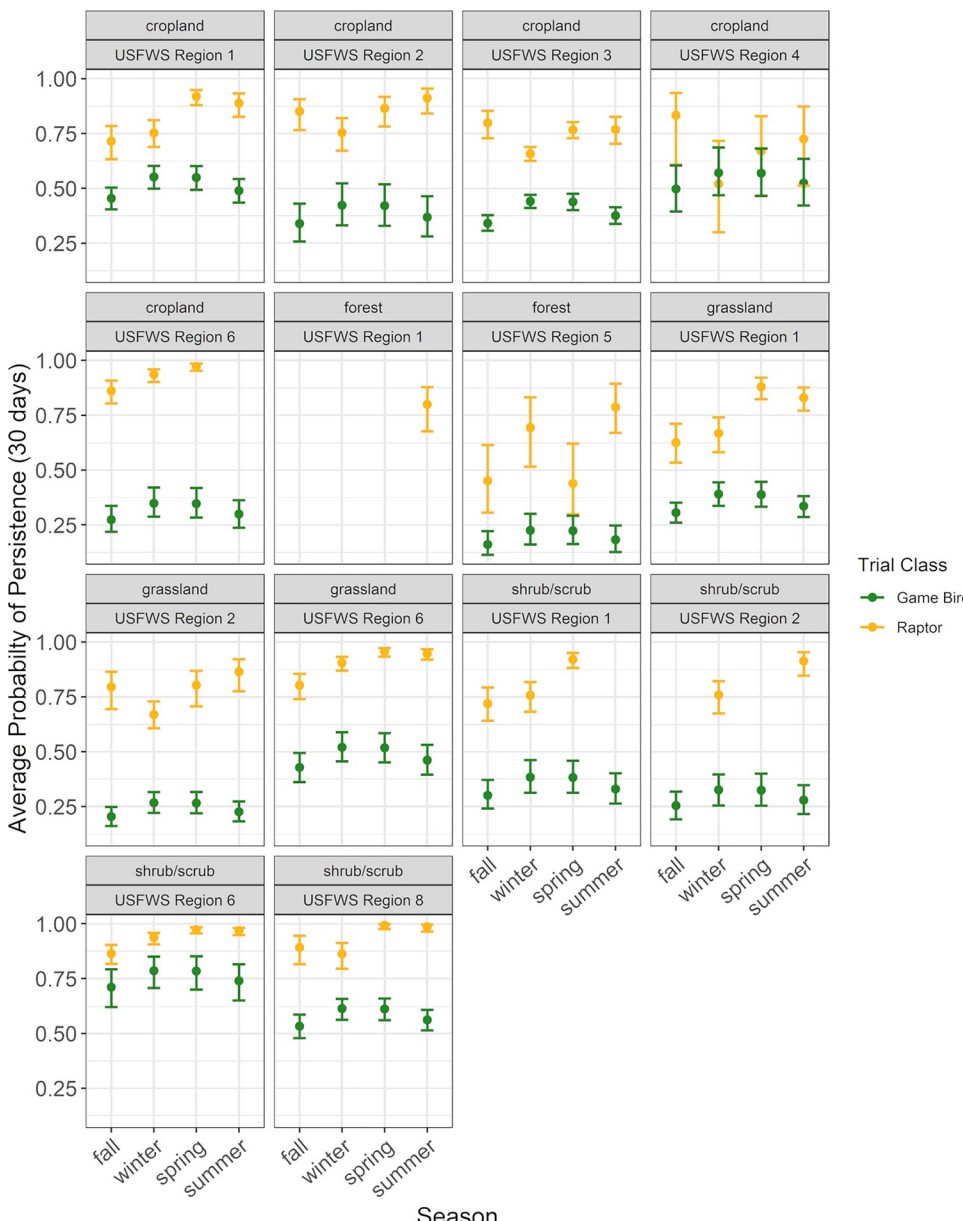

**Fig 10. Modeled average probability of persistence through a 30-day search interval for all raptor and game bird carcass meta-data, limited to sampled strata.** Average probability of persistence through 60- and 90-day search intervals for both bird types followed the patterns shown here, but were shifted lower on the vertical axis because average probabilities of persistence decreased as search intervals increased.

S7 and S8 Tables). Raptor carcasses consistently persisted longer than 66 days (with median persistence times ranging from 11 to 604 days), while game bird carcasses consistently persisted fewer than 8 days (ranging from 2 to 48 days). Furthermore, the 90% CIs for median persistence did not overlap for 58 out of 61 strata combinations, with the only overlap between the bird types occurring within 3 strata combinations: cropland habitat in Region 4 during winter, spring, and summer (S7 and S8 Tables). Thus, raptor persistence was significantly higher ($\alpha = 0.10$) than game bird persistence for 95% of the sampled strata. The 90% CIs for probabilities of persistence through 30-, 60-, and 90-day search intervals overlapped for the

**Table 3. Model selection results using AICc for linear mixed-effects models of logit transformed raptor average probability of persistence, based on 652,000 simulated persistence values and 27 studies.**

| Model Formula | AICc | Δ AICc |
|---|---|---|
| logit(Raptor)~ Gamebird+Season+ Region+Habitat + (1 + Gamebird \| Analysis Group) | 1045454.04 | 0[a] |
| logit(Raptor)~ Gamebird+Season + (1 + Gamebird \| Analysis Group) | 1045457.70 | 3.66 |
| logit(Raptor)~ Gamebird+Season+Habitat + (1 + Gamebird \| Analysis Group) | 1045458.27 | 4.23 |
| logit(Raptor)~ Gamebird+Season+Region + (1 + Gamebird \| Analysis Group) | 1045461.57 | 7.53 |

Models with Δ AICc (difference in AIC points from top model) less than or equal to 10 are shown above.

[a] We used this model in the analysis.

same 3 strata combinations, as well as for 60- and 90-day search intervals in forest habitat in Region 5 during spring (Fig 10, S7 and S8 Tables). Variability in 90% CIs was highest in raptor persistence estimates for Regions 4 and 5 (Fig 10, S9 Table).

**Scaling game bird persistence.** To meet Objective 3, the filtered meta-dataset resulted in fitting 118 sets of interval-censored survival regression models to analysis groups with both game bird and raptor persistence data (see S11 Table for model selection results). The 652,000 bootstrapped average probability of persistence pairs (1,000 bootstrap replicates for each combination of strata in the analysis-group specific models and search intervals of 14, 30, 60, and 90 days) from each study were used to fit 32 candidate LMEMs and calculate AICc (models with Δ AICc less than or equal to 10 are shown in Table 3). Due to model convergence failures, all data from the Region 5 forested Study Site had to be removed from consideration because these data were the only instance of forest habitat or Region 5 in the dataset (in contrast, these data were used in the model validation step, below). A potential effect of inter-annual variability would have been captured in the residual error term of the best-supported model; given the favorable validation results (below), we do not believe the predictions of the model are systematically affected by annual variation.

The best-supported model of logit transformed average probability of persistence for raptors included game bird average probability of persistence, season, Region, and habitat as fixed effects, and a random intercept and slope by analysis group; parameter estimates of the selected model are provided in Table 4 and predictions are visualized in Fig 11. Because the categorical covariates (season, Region, and habitat) had to be converted into indicator variables during LMEM fitting, 1 level of each covariate was absorbed in the intercept as "baseline" levels. The baseline levels of the categorical variables in the top model were fall, Region 1, and cropland for season, Region, and Habitat, respectively. The selected model had a strong positive relationship between raptor and game bird average probability of persistence, which was both highly significant (p-value < 0.001; Table 3) and had the lowest coefficient of variation (SE / estimate) among the parameters (treating SE < 0.01 as 0.01 for the calculation of the coefficient of variation; Table 3), while accounting for effects of season, Region, and habitat. When controlling for other factors, raptor persistence probabilities were highest in Region 8 (coefficient of 2.86, SE of 0.84) and lowest in Region 4 (coefficient of -0.39, SE of 0.73) Based on the signs of the remaining categorical variables, winter, spring, and summer decreased predicted raptor average probability of persistence slightly relative to fall. Grassland habitat had a modestly positive effect (coefficient of 0.39, SE of 0.41) on raptor average probability of persistence while shrub/scrub habitat had a negative effect (coefficient of -2.20, SE of 0.78) compared to cropland habitat.

The magnitude of the random effects due to analysis groups on the model slope (SE = 2.32) and intercept (SE = 0.53) were both smaller than the corresponding fixed effect parameters

**Table 4. Parameter estimates for best supported (AICc) linear mixed-effects model of raptor average probability of persistence as a function of game bird average probability of persistence.**

| Fixed Effects | Estimates | SE | p-value |
|---|---|---|---|
| (Intercept; incorporates Region 1, fall and cropland habitat categories) | -0.66 | 0.38 | 0.085 |
| Game bird average probability of persistence | 4.19 | 0.29 | <0.001 |
| Season [spring] | -0.05 | <0.01 | <0.001 |
| Season [summer] | -0.03 | <0.01 | <0.001 |
| Season [winter] | -0.08 | <0.01 | <0.001 |
| Region [2] | 0.47 | 0.46 | 0.301 |
| Region [3] | 0.33 | 0.43 | 0.439 |
| Region [4] | -0.39 | 0.73 | 0.597 |
| Region [6] | 1.25 | 0.49 | 0.010 |
| Region [8] | 2.86 | 0.84 | 0.001 |
| Habitat [grassland] | 0.39 | 0.41 | 0.332 |
| Habitat [shrub/scrub] | -2.20 | 0.78 | 0.005 |
| **Random Effects** | | **SE** | |
| Residual error | | 0.29 | |
| Analysis group (intercept) | | 0.53 | |
| Analysis group x game bird (slope) | | 2.32 | |
| Intra-class correlation coefficient | | 0.66 | |
| n Analysis group | | 27 | |

(4.19 and -0.66, respectively; Table 4). The intra-class correlation coefficient was 0.66, which indicates a moderate degree of association between the average probability of persistence estimates generated from the same analysis group; in other words, approximately 66% of the variability in logit-transformed raptor average probability of persistence not explained by the model is attributable to the variability in the average probability of persistence estimates coming from each analysis group.

Results of model validation showed consistently high positive correlation (minimum of 0.72, maximum of 0.89) between predicted raptor average probability of persistence and the actual raptor average probability of persistence estimates from individual analysis groups in each hold out set (Table 5). Similarly, RMSE and average absolute error were both consistent across the validation sets, ranging from 0.11 to 0.16, and 0.08 to 0.13, respectively. The range of RMSE resulting from the validation sets is indicative of the expected error in a new prediction, meaning we would expect a new prediction of raptor average probability of persistence from the LMEM to be within 0.11–0.16 of the raptor average probability of persistence estimated from the site-specific data alone. The direction of errors was most variable among the 4 validation metrics, with the proportion of predictions exceeding the true estimate ranging from 0.25 to 0.90 (median of 0.67), suggesting predictions may be more likely to err higher than lower. However, the consistently low RMSE and average absolute error suggest model predictions would rarely overestimate by more than 0.16.

## Discussion

Our results indicated that raptor carcasses persisted significantly longer than game bird carcasses in both the Field Trial Analyses and the Raptor and Game Bird Persistence Meta-analysis, with median raptor persistence more than doubling median game bird persistence in nearly every strata combination of season, Region, and habitat for which we had data. Thus, the probability of persistence for game birds will be lower than raptors, thereby leading to a

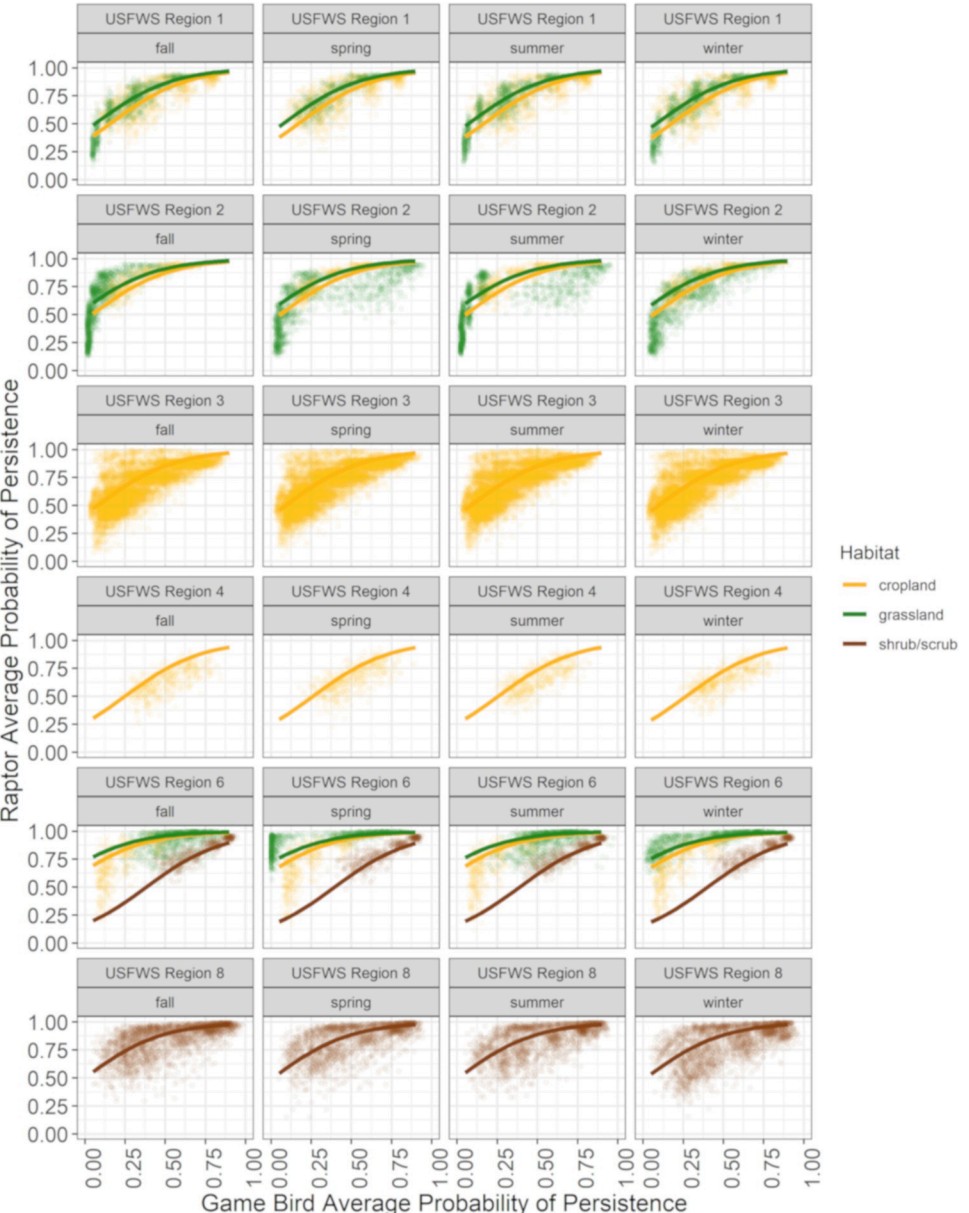

**Fig 11. Average probability of persistence for game birds and raptors (points) based on bootstrapped data from 118 study-specific interval-censored survival regression models, with predictions of the best supported linear mixed-effects model (lines), by season, Region, and habitat.** Estimates of average probability of persistence were calculated for 14-, 30-, 60-, and 90-day intervals to provide a range of plausible persistence probabilities for post-construction fatality monitoring.

higher raptor fatality estimate if unadjusted game bird persistence metrics are used. However, logistical challenges could prohibit the use of raptor trials at many wind facilities. We developed linear mixed-effects models that predict raptor persistence probabilities based on estimated game bird persistence probabilities. Our scaling model provides a practical statistical tool to address gaps in raptor persistence data at sites in a broad range of landscape contexts in the continental U.S., and should be used to inform fatality estimation when site-specific raptor persistence data are limited or absent.

**Table 5. Validation metrics for 10 random cross-validation sets.** For each validation set the following metrics were calculated: RMSE, Pearson's correlation coefficient, proportion of predictions exceeding the OOS estimates, and average absolute error (average of the absolute value of the difference between predicted and actual raptor average probability of persistence).

| RMSE | Correlation (Pearson's) | Proportion of Predictions > OOS Estimate | Average Absolute Error $\lvert \widehat{r}_{actual} - \widehat{r}_{prediction} \rvert$ |
|---|---|---|---|
| 0.12 | 0.79 | 0.52 | 0.08 |
| 0.12 | 0.83 | 0.25 | 0.09 |
| 0.11 | 0.85 | 0.82 | 0.09 |
| 0.12 | 0.79 | 0.77 | 0.11 |
| 0.14 | 0.72 | 0.58 | 0.09 |
| 0.16 | 0.89 | 0.91 | 0.13 |
| 0.11 | 0.78 | 0.64 | 0.09 |
| 0.11 | 0.79 | 0.70 | 0.09 |
| 0.11 | 0.88 | 0.82 | 0.09 |
| 0.11 | 0.80 | 0.58 | 0.08 |

## The importance of accurate persistence estimates in eagle fatality monitoring

Overestimating eagle fatalities due to the use of inappropriate surrogates in CPT can have major ramifications for wind facility operators, transmission line utilities, and other entities conducting PCFM. As estimated carcass persistence times shorten, the bias adjustment becomes larger and the resulting fatality rate estimates increase, at times substantially, relative to the number of actual carcasses found during searches. For eagle incidental take permit holders, artificially inflated fatality estimates could have significant ramifications over the life of a permit term in 3 ways. First, overestimated fatality rates could be interpreted as a facility being on a trajectory to exceed permitted take over the permit term. The perceived risk of falling out of permit compliance may trigger the implementation of costly, and unnecessary, adaptive management measures (e.g., increased fatality monitoring effort, turbine curtailment, installation of detect/deter technologies). Second, permitted take may be interpreted as being exceeded even after the implementation of adaptive management measures, leading to permittees being deemed out of compliance with their permit terms. Falling out of permit compliance may result in permit suspension or revocation, and the U.S. Department of Justice or the USFWS's Office of Law Enforcement may determine enforcement action in response to unauthorized take is warranted [44]. Third, an overestimate of eagle fatalities may result in increased mitigation requirements and costs to offset the inflated take estimate. Due to the importance of accurate persistence estimates in PCFM, we recommend incorporating adjusted (i.e., scaled) game bird persistence data, as supported by our scaling model and described below, when estimating fatality rates for eagles and other large raptor species. When CPT are necessary at a facility, they should be completed with carcasses as similar to the species detected as fatalities (e.g., large raptors for eagle fatality monitoring), or for which fatality rates are being estimated, as much as possible.

Accurate eagle and raptor fatality estimates would be advantageous to wildlife agencies and researchers aiming to understand population size, population trends, and population level impacts of mortality. Recent studies have focused on how installed wind energy, and future build-out, could affect population trends of raptors [45–47]. However, an acknowledged deficiency of the modeling approaches used in these assessments is that bias correction data for raptors is often aggregated with other large birds (e.g., game birds), making raptor-specific fatality estimates of limited value. For example, Diffendorfer et al. [46] stated they were unable

to apply species-specific carcass persistence adjustments to detected carcasses in their analysis and thus did not account for this factor in their fatality estimates. Our results indicate the average probability of persistence for a 30-day search interval was generally high for raptors, but as low as 0.46 in 1 habitat × season combination (forest during spring). By not accounting for persistence time and essentially assuming the probability of persistence was 1.0, it is likely Diffendorfer et al. [46] underestimated fatality rates in their analysis. Given the interest in population responses to renewable energy development [47], our results provide a starting point for understanding raptor persistence times on a broad scale, and as the collective dataset expands, species-specific persistence times could become an option for improved accuracy in fatality estimates and corresponding population impact assessments.

Although persistence probabilities generated from large raptor carcasses or scaled game bird carcasses are a substantial improvement over game bird surrogates alone, the resulting estimates are likely conservative when adjusting for bias in eagle fatality estimates. Scavenging, and thus carcass persistence, can also be influenced by carcass size [48–53]. An analysis of data from 44 carcass persistence studies found that the larger the carcass, the longer the carcass persisted [3]. Santos et al. [51] found that road-killed birds with higher body masses had higher persistence probabilities, possibly due to smaller carcasses having a wider range of potential scavengers than larger species. Bernardino et al. [5] found that larger carcasses were not totally removed as quickly as smaller carcasses, and the remains also persisted longer in the field. In our study, we set a minimum "large" raptor size threshold of approximately 30-cm wing chord and 300-g mass; however, even the largest carcasses included in our study (black vulture [40-cm wing chord, 2,150-g mass, n = 1 carcass] and turkey vulture [51-cm wing chord, 2,000-g mass, n = 61 carcasses]), were smaller in size and less than half the mass than the average bald eagle (*Haliaeetus leucocephalu*s; 61-cm wing chord, 4,900-g mass) or golden eagle (*Aquila chrysaetos*; 59-cm wing chord, 4,000-g mass; measurements from Cornell Lab of Ornithology [54]). Data on eagle carcass persistence are limited, but a study in Utah and Colorado documented an average probability of persistence for a 30-day search interval of 0.97 for eagle carcasses and 0.76 for other large raptor carcasses [55]. Therefore, eagles should be even more difficult for common scavenger species to completely remove (i.e., drag away or consume) compared to the carcasses we used in this study. Using persistence times of actual eagle carcasses, were it possible, would very likely further reduce the bias adjustment in eagle fatality estimation and lower the resulting take estimates.

### Explaining patterns in raptor and game bird carcass persistence

Our results are consistent with results from previous studies documenting extended persistence times exhibited by raptor carcasses [6–9, 13], and several causal mechanisms could underlie this phenomenon. Carcasses of predator species, such as raptors, may generally be less available on the landscape and are outside of the typical "search image" and scavenging preferences exhibited by scavenging species. Avoidance of predator or carnivore carcasses by scavengers was noted by Robertson [56], who suggested that properties intrinsic to a carcass (e.g., coloration) can influence scavenger dynamics. Moleón et al. [52] found that the mean number of species observed feeding at herbivore carcasses were substantially higher than at carnivore carcasses; the authors interpreted their results along with findings in Selva et al. [57] and Olson et al. [58] as avoidance behavior, speculating this behavior may be driven by disease-induced mortality (e.g., transmission of parasites) associated with phylogenetically similar species. Alternatively, longer raptor persistence times may be due to the fact that raptor carcasses are less likely to be fully scavenged. Studies have found that raptor carcasses persisted longer before being initially scavenged, but also were less likely to be completely removed than

non-raptor counterparts [8]. This was also true in our study, where 81.0% of game bird carcasses were completely removed prior to the end of the trial, compared to 30.8% of raptor carcasses. Of CPT monitored by camera in our study, coyotes (*Canis latrans*) were the most commonly recorded scavenger; coyotes removed 13 of 18 game bird carcasses visited, but only removed 4 of 12 raptor carcasses visited.

Temperature and season have been associated with scavenging rates and decomposition in other persistence studies. Guinard et al. [59] found that persistence probabilities for barn owls were highest during summer, consistent with our Field Trial Analyses results. Decomposers are known to compete with scavengers for carcasses [48]. Janzen [60] found that microbes utilizing carcasses produce toxins that are dangerous to most vertebrates. At high concentrations, such as may occur during high summer temperatures [61], compounds produced by these microbes likely repel vertebrate scavengers. Microbes, however, do not fully consume carcasses; after soft body parts of raptors have been consumed, high temperatures and dry conditions can lead to desiccation of remains, which then persist for relatively long times [51]. Desiccation is likely more prevalent during the summer and early fall, when we found persistence to be longer than in other seasons. Arid, open environments, such as exist throughout the majority of Regions 2, 6, and 8, are also going to be conducive to desiccation of carcass remains. Regions 1 and 4, conversely, generally have higher humidity and lower average temperatures, which may be tied to the shorter persistence we documented in these Regions. The decomposition process may also have caused the higher variability in median raptor persistence time during our summer CPT, especially in grassland and shrub/scrub habitats.

Increased variability in raptor persistence during the summer season was driven by a higher proportion of carcasses remaining intact until the end of the CPT period (i.e., right-censored trial data). When there are more right-censored data in the model, the resulting predictions tend to be more variable on the high side of the estimate, as the model lacks information about how much beyond the CPT length a carcass persisted. A high frequency of right-censored data also tends to result in median persistence time estimates that exceed the maximum trial length used to monitor carcasses. Although median persistence times approaching a year or more may not be biologically relevant, these estimates show a researcher can comfortably rely on an average carcass to persist through the much shorter search intervals commonly used during PCFM.

A potential temporal mechanism that may drive the lower raptor average probability of persistence we observed during winter could be the increased reliance of some facultative scavengers on scavenging during this season, when predation becomes more difficult [62]. Facultative scavengers have been shown to rely more heavily on predation than scavenging during summer, when prey are more abundant [63]. The shift in foraging decision-making may be more pronounced during the longer, colder periods endured within northern Regions (e.g., Region 1, Region 4). Contrary to our findings, some studies have documented longer persistence times during cold and/or rainy seasons [64, 65]. In some cases, carcass detection may be easier during warmer and/or drier seasons, particularly if snowfall interferes with detection by blocking olfactory or visual cues. Further study is needed to better understand how seasonal effects influence raptor carcass persistence under different environmental conditions.

Habitat has also been identified as an important driver of carcass persistence in several other studies [48, 51, 66, 67], but did not significantly influence persistence times in others [3, 65, 68]. In our analysis of the full meta-dataset, median persistence times of both bird types were highest in shrub/scrub habitat, followed by cropland, grassland, and forest habitats (S9 and S10 Tables). Shrub/scrub habitat is found in arid, open environments, which may prolong persistence through processes, similar to those described above, such as potential seasonal and geographic mechanisms influencing persistence times. Cropland habitat is generally barren

outside of the growing season, which may also support relatively rapid desiccation of the carcasses, or limit scavenger activity due to lack of cover. Conversely, forest habitat has been shown to support increased species richness in scavenger guilds, which supports the prevalence of facultative scavenging [57]; forest habitat can also provide cooler, more humid microclimates known to facilitate more rapid carcass decomposition rates [49]. Our study included limited CPT data within forest habitat: it is difficult to interpret broad-scale effects of forest habitat on persistence without additional raptor and game bird CPT data within forest habitat across multiple Regions.

## Predicting raptor persistence at a wind facility using game birds

Our model to scale game bird persistence provides a practical statistical method to address gaps in raptor persistence data at sites in a broad range of landscape contexts in the continental U.S. The linear mixed-effects scaling model is capable of predicting raptor persistence as a function of game bird persistence, accounting for effects of season, Region, habitat, and variability in site to site measures of raptor and game bird persistence. Our model showed strong predictive power (Pearson's correlation 0.72 to 0.89) and low error during model validation (RMSE 0.11 to 0.16). The favorable validation results also suggest our decision to leave out interaction terms and a trend (year) covariate were not overly detrimental to predicting raptor average probability of persistence. We found a strong positive relationship between raptor and game bird average probability of persistence. Fatality estimates for eagles and other large raptor species that incorporate unadjusted game bird persistence data will therefore be biased high, with ramifications as described above.

Predictions of raptor persistence based on scaled game bird data can be accomplished in a variety of PCFM scenarios for eagles or large raptors. In the simplest application, the scaling model can be used to adjust site-specific game bird data at site that falls within 1 of the Regions, habitats, and seasons included in our model. Many eagle incidental take permit holders may only have persistence data for game bird carcasses available, as traditional methods have resulted in game bird carcasses being used for site-specific CPT in most instances, and raptor carcasses remain difficult to obtain. In a scenario where game bird persistence data are available and raptor persistence data are not, our scaling model can be used to predict a raptor average probability of persistence for any search interval of interest, up to 90 days. For example, the 30-day game bird average probability of persistence for 1 study (included in the meta-dataset) in Region 8 shrub/scrub habitat was 0.60 during summer; our scaling model resulted in a predicted 30-day raptor average probability of persistence of 0.85 (for reference, the actual raptor average probability of persistence was 0.84 based on site-specific data). A researcher would therefore use the 0.85 average probability of persistence when calculating fatality estimates for eagles and other large raptors at the project, resulting in more accurate, and lower, take estimates.

For sites in a habitat and/or Region for which we did not have or include data (e.g., a site in Region 5 or a site in forest habitat in any Region), researchers will need to weigh the assumptions being made by using (for example) game bird average probability of persistence from a nearby site in different habitat and/or a different Region for scaling up to raptor average probability of persistence. The meta-analysis showed greater variability in raptor and game bird average probability of persistence in some strata (e.g., cropland in Region 4, forest in Region 5; Fig 10) compared to others; however, our meta-analysis and model show scaling game bird to raptor average probability of persistence under even a highly conservative assumption (e.g., cropland in Region 4) would result in a more accurate estimate of raptor persistence than an estimate of game bird persistence in effectively every case.

Some applicants will not have any game bird or raptor persistence data available for use in fatality estimation. In a scenario where there is a complete absence of persistence data at a project of interest, a researcher could use an estimate of raptor average probability of persistence from a nearby project (same Region) in similar habitat if data were available. If recent raptor persistence data were lacking at a regional scale, game bird average probability of persistence from a nearby project in similar habitat, or from our Game Bird Persistence Meta-analysis could be used as input to the scaling model to develop a raptor average probability of persistence. As an example, we can take the minimum of the lower 90% CI bounds of seasonal 30-day game bird average probability of persistence for a facility in Region 3 in cropland habitat (fall, 0.31; S10 Table). Using game bird average probability of persistence of 0.31, the scaling model would predict a raptor average probability of persistence of 0.72 for a facility in Region 3 cropland habitat. Given the estimated raptor average probability of persistence for Region 3 cropland habitat ranged from 0.66 (winter) to 0.80 (fall), a prediction of 0.72 would be a reasonably conservative estimate of 30-day raptor average probability of persistence.

Researchers can also use the information provided above for an a priori determination of an appropriate search interval to meet a desirable overall probability of detection at their facility. Permittees hope to demonstrate take permit compliance while also using optimized search protocols that balance detection probability with monitoring effort and costs. Our study results can be used to forecast persistence times for large raptors with confidence under many conditions throughout the continental U.S. Average probabilities of persistence can be calculated for bi-weekly, monthly, quarterly, or other search intervals of interest.

## Summary and suggestions for future research

Overestimating eagle and other large raptor fatalities due to the use of inappropriate surrogates in CPT can have major ramifications for entities conducting PCFM. As estimated carcass persistence probabilities are reduced, the bias adjustment becomes larger and the resulting fatality rate estimates can increase substantially. Our results show that large raptor carcasses persisted significantly longer than game bird carcasses; therefore, eagle and other large raptor fatality rates incorporating unadjusted game bird persistence data collected carcasses will be inflated. Our study, using the most comprehensive CPT database available, documented predictable patterns in large raptor persistence: carcasses persisted longer in dry, warm seasons and in arid, open habitats and landscapes. In the absence of species-specific persistence data (e.g., eagles), researchers should use reasonable surrogates (e.g., large raptors) for CPT whenever possible to measure persistence bias and thereby improve the accuracy of the resulting fatality estimates. As large raptor carcasses are not always available—particularly as demand increases —our study provides a solution: a practical statistical model for scaling game bird persistence data upwards to predict large raptor persistence. Even in the absence of any persistence data, results from this study can be used to more accurately estimate eagle and other large raptor fatality rates using a large raptor probability of persistence prediction modeled from game bird CPT data collected at projects sited in similar conditions. Although our results show complex influences among Region, habitat, and season, and other factors can influence persistence times, we found consistent and repeatable patterns in Raptor and Game Bird Persistence Meta-analysis and suggest these patterns are likely evident in all areas with similar scavenger communities. Additional CPT data from combinations of strata for which we had no data or limited data would strengthen our model and lead to broader applicability in estimating raptor persistence using new or existing data from game bird CPT at facilities in these areas. Furthermore, the development of a simple analysis tool (e.g., an R package or application) incorporating our scaling model would enable researchers to scale existing game bird persistence

probabilities to raptor persistence probabilities with confidence intervals, allowing more accurate fatality estimates for raptors and eagles to be calculated across a wide range of projects.

## Supporting information

**S1 Table. Full meta-analysis dataset.** Table includes all raw data from the data curation effort (in addition to the data collected during our Field Trials). Trials are aggregated by study (the trial placement effort during a period up to a year, at a single facility), predominant habitat at the facility, Region, and season.
(DOCX)

**S2 Table. Field trial breakdown.** Summary of the carcasses (240 game birds and 239 raptors) used in field trials at the six Study Sites.
(DOCX)

**S3 Table. Raptor persistence model selection.** Model selection used corrected Akaike's Information Criterion (AICc) with seasonal and habitat covariates for the 239 large raptor carcasses placed during the carcass persistence study conducted June 2020 –August 2021.
(DOCX)

**S4 Table. Estimates of median raptor carcass persistence times (in days) and average probabilities of persistence.** Median persistence times and probabilities of persistence are for 3 search intervals (SIs; 30 days, 60 days, and 90 days), with 90% confidence intervals (CIs), for the carcass persistence study conducted from June 2020 –August 2021.
(DOCX)

**S5 Table. Game bird persistence model selection.** Model selection used corrected Akaike's Information Criterion (AICc) with seasonal and habitat covariates for the 240 large game bird carcasses placed during the carcass persistence study conducted June 2020 –August 2021.
(DOCX)

**S6 Table. Estimates of median game bird persistence times (in days) and average probabilities of persistence.** Median persistence times and probabilities of persistence are for 3 search intervals (SIs; 30 days, 60 days, and 90 days), and 90% confidence intervals (CIs), for the carcass persistence study conducted from June 2020 –August 2021.
(DOCX)

**S7 Table. Raptor persistence model selection.** Model selection used corrected Akaike's Information Criterion (AICc) with seasonal, habitat, and USFWS Region covariates for the meta-dataset collected for the carcass persistence study.
(DOCX)

**S8 Table. Game bird persistence model selection.** Model selection used corrected Akaike's Information Criterion (AICc) with seasonal, habitat, and USFWS Region covariates for the meta-dataset collected for the carcass persistence study.
(DOCX)

**S9 Table. Estimates of median raptor persistence times and average probabilities of persistence.** Median persistence times and average probabilities of persistence are by USFWS Region, habitat, and season for 3 search intervals (30 days, 60 days, and 90 days), with 90% confidence intervals (CIs).
(DOCX)

**S10 Table. Estimates of median game bird persistence times and average probabilities of persistence.** Median persistence times and average probabilities of persistence are by USFWS Region, habitat, and season for 3 search intervals (30 days, 60 days, and 90 days), with 90% confidence intervals (CIs).
(DOCX)

**S11 Table. Model selection results using interval-censored survival regression to scale game bird persistence.**
(DOCX)

## Acknowledgments

Will Vesely and Juan Botero served as the Renewable Energy Wildlife Research Fund's (REWRF) project managers and provided valuable administrative support during our research. We would like to thank the Operations and Maintenance staff members at our 6 Study Sites for providing safety instruction, access, storage, and other logistical considerations for the study. Andrea Palochak and David Klein (both with WEST) provided valuable assistance during manuscript preparation. Christopher Murray and Faith Kulzer (both with WEST) provided support organizing, formatting, and preparing data for analyses. We thank Joel Thompson (with WEST) who offered thoughtful comments on a draft manuscript. Many members of the REWRF provided valuable raptor persistence data for curation of the meta-dataset. Lastly, we thank an anonymous reviewer who offered thoughtful comments on our initial manuscript submittal.

## Author Contributions

**Conceptualization:** Eric Hallingstad, Daniel Riser-Espinoza.

**Data curation:** Daniel Riser-Espinoza, Samantha Brown.

**Formal analysis:** Daniel Riser-Espinoza.

**Funding acquisition:** Eric Hallingstad.

**Investigation:** Eric Hallingstad, Daniel Riser-Espinoza, Samantha Brown.

**Methodology:** Eric Hallingstad, Daniel Riser-Espinoza, Samantha Brown.

**Project administration:** Eric Hallingstad.

**Resources:** Eric Hallingstad, Samantha Brown, Jeanette Haddock.

**Supervision:** Eric Hallingstad.

**Validation:** Daniel Riser-Espinoza.

**Visualization:** Eric Hallingstad, Daniel Riser-Espinoza, Paul Rabie, Karl Kosciuch.

**Writing – original draft:** Eric Hallingstad, Daniel Riser-Espinoza, Samantha Brown.

**Writing – review & editing:** Eric Hallingstad, Daniel Riser-Espinoza, Paul Rabie, Karl Kosciuch.

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
