## [Decision Letter · Decision Letter 0]

2 Nov 2022

PONE-D-22-26139GAME BIRD CARCASSES ARE LESS PERSISTENT THAN RAPTOR CARCASSES, BUT CAN PREDICT RAPTOR PERSISTENCE DYNAMICSPLOS ONE

Dear Dr. Hallingstad, Thank you for submitting your manuscript to PLOS ONE. After careful consideration, we feel that it has merit but does not fully meet PLOS ONE’s publication criteria as it currently stands. Therefore, we invite you to submit a revised version of the manuscript that addresses the points raised during the review process.

Dear Dr. Eric Hallingstad,

Thank you for submitting your study to PLOS ONE.

Since late September, I requested 15 experts to review your submission PONE-D-22-26139.

Most of them were not avialable due to time shortage or conflicts of interest.

However, I received a constructive and excellent review from an expert in this research field.

I also read your submission and I agree with this reviewer - you have written and excelent manuscript based on a high quality research. Thus, I ask that you make changes to improve its quality even more, and publish a new (shorter) version in PLOS ONE.

Although there is NO LIMIT for manuscript length in PLOS ONE, I consider that your manuscript is too long. The reviewer has the same opinion.

The reading was a bit difficult due to this.

I suggest that you try to reduce the number of tables and figures, keeping only the major results, and the corresponding paragraphs.

As an expert in this research field, the reviewer suggested you to keep with the 3 objectives, and some additional minor suggestions also were provided. I suggest that you try to follow these major and minor suggestions. Please see below.

Due to the high quality and constructive review, I consider that we can advance with the evaluation process with only this review (what is allowed in this journal).

I ask that you check carefully the new numbers of figures and tables, and the corresponding text prior to submitting a corrected version.

Please feel free to contact me if you need additional time to provide a corrected version.

There are additional PLOS ONE instructions below that you might also consider. 

Dárius Tubelis, Ph.D.

We look forward to receiving your revised manuscript.

Kind regards,

Dárius Pukenis Tubelis, Ph.D.

Academic Editor

PLOS ONE

Journal Requirements:

"Will Vesely and Juan Botero served as the Renewable Energy Wildlife Research Fund’s (REWRF) project managers and provided valuable administrative support during our research.. This study was funded by REWRF."

"Funding for this research was provided to the authors (WEST, Inc.) by the Renewable Energy Wildlife Research Fund (REWRF; https://rewi.org/2022/03/24/research-fund-expands-to-solar-becomes-renewable-energy-wildlife-research-fund/). The funders had no role in data collection and analysis, or preparation of the manuscript. REWRF members reviewed the manuscript, approved the use of their existing data, and participated in the decision to publish."

"I (EH) have read the journal's policy and the authors of this manuscript have the following competing interests: funding for the research was provided by the Renewable Energy Wildlife Research Fund. REWRF members reviewed the manuscript, approved the use of their existing data, and participated in the decision to publish. All authors work for Western EcoSystems Technology, Inc., an environmental consulting firm. There are no patents, products in development, or marketed products to declare. The above disclosures do not alter the authors' adherence to all the PLOS ONE policies on sharing data and materials, as detailed online in the guide for authors."

5. We note that Figures 1, 3, 8 and 9 in your submission contain map images which may be copyrighted. All PLOS content is published under the Creative Commons Attribution License (CC BY 4.0), which means that the manuscript, images, and Supporting Information files will be freely available online, and any third party is permitted to access, download, copy, distribute, and use these materials in any way, even commercially, with proper attribution. For these reasons, we cannot publish previously copyrighted maps or satellite images created using proprietary data, such as Google software (Google Maps, Street View, and Earth). For more information, see our copyright guidelines: http://journals.plos.org/plosone/s/licenses-and-copyright.

(1) You may seek permission from the original copyright holder of Figures 1, 3, 8 and 9 to publish the content specifically under the CC BY 4.0 license.   

Reviewers' comments:

Reviewer's Responses to Questions

**Comments to the Author**

1. Is the manuscript technically sound, and do the data support the conclusions?

Reviewer #1: Yes

2. Has the statistical analysis been performed appropriately and rigorously? 

Reviewer #1: Yes

3. Have the authors made all data underlying the findings in their manuscript fully available?

Reviewer #1: Yes

4. Is the manuscript presented in an intelligible fashion and written in standard English?

Reviewer #1: Yes

5. Review Comments to the Author

Reviewer #1: This is an interesting and well-written study that addresses a topic still understudied: the bias created by used of game bird carcasses (instead of wild specimens) in carcass persistence trials, and their impact on the accuracy of the estimates of bird fatality (more specifically of raptors) in post-construction monitoring studies. In addition to a meta-analyis, the authors created a mixed-effects model that predicts raptor persistence as a function of game bird persistence, which will be of great value for monitoring studies (in US) where raptor persistence data is absent.

I have, however, some reservations about the length of the manuscript (> 10,000 words, excluding tables) and the way the results are presented (looking more like a report than a research paper). For instance, if objective 1 is to “compare persistence times and probabilities for large raptor and game bird…”, why are not the results focused on that (3rd section of the “Field trial results”)? Is it necessary to describe separately and in such detail the persistence of raptors and game birds? The same for objective 2: raptor and game bird persistence really need to be described separately, or the focus should be on the “Patterns in raptor and game bird persistence”?

Another problem is the numerous tables included in manuscript, with some of the figures showing the exact same information (for details see my specific comments below).

In summary, and although presenting the methods and results in a descriptive way is generally regarded as a positive thing, here it makes the manuscript too long and hard to read, which ultimately diverts the readers’ attention from the results obtained for the 3 objectives set for the study.

Specific comments:

L. 35 – Unless you have read the paper, it’s not clear to what meta-dataset you are referring to.

L. 63 – I would say that it the opposite: “if the bias parameters are high”, meaning if the number of missed carcasses (by imperfect detection or removal), the no. of carcasses that fall outside or the no. of unsearched turbines are high, then the difference between the no. of carcasses found and the no. of fatalities estimated is greater. It would be “low” only if it had been previously explained (L. 60-62) that the bias-correction parameters (by which the observed fatality is divided) are the carcass persistence probability, the detection rate (by the observers), the proportion of carcasses that fall within the search area, and the proportion of searched turbines.

L. 90-91 – There has not been a through investigation of persistence data from raptors only, or game vs. wild specimens… But for birds in general there are several studies that have compiled and investigated persistence data from more than a single site, or a small number of sites (e.g., Barrientos et al 2018; Smallwood 2013, doi:10.1002/wsb.260; Bispo et al. 2013, doi: 10.1007/s10651-012-0212-5).

Figure 1 and 3 - Image resolution isn't high enough (it looks pixelated).

L. 176 – The use of the word “fresh” here may be misleading since all carcasses were frozen and then thawed before the trials.

L. 200-202 / 207-209 – Include reference or explain why this threshold was used.

L. 221 – Not sure why the data from cameratraps had to be “consistent with the in-person checks”, if the R-package “GenEst” was used (see comment to L. 261-276). GenEst can accommodate data both from weekly visits (in-person checks) and from smaller time intervals (cameratraps), as long as a “LastPresent” time and “FirstAbsent” time if provided for each carcass.

L. 257 – Replace “or” by “of”.

L. 261-276 / 316-330 – Were theses analyses done using the GenEst R-package [33]? It is not clear since Ref. 31 to 34 are always cited together, while for the 3rd analysis (“scaling game bird persistence”) it is clearly stated.

L. 378 – Start of stage 2?

L. 396-397 - This last step of stage 2 is not represented in Figure 3, is it?

Tables 3, 5, 7 and 8 – To reduce the extended number of tables in the MS, please consider the possibility of moving these 4 tables to Supplementary material, as the best supported models for raptor and game persistence are stated in the text, both in objective 1 and 2.

Table 4 and 6 / Figure 4 to 7 – These tables and figures show the exact same information, right? I would suggest moving Tables 4 and 6 to Supplementary material, with the estimated values still being available for those who may need it.

Tables 9 and 10 – These tables are too long and, consequently, difficult to read… Please considerer moving them to Supplementary material, and better explaining (L. 582-87) why you used the 30-day interval as a showcase your overall results (Fig. 10).

L. 522 – I suggest turning this section into a 2nd-level heading, for a better correspondence between the three study objectives (established in the Introduction section) and the structure of the Results section.

L. 763-766 – The persistence of road-kills is highly influenced by traffic; thus, it is not comparable with carcass persistence dynamics at WEFs and power lines, where removal occurs mostly due to scavenging or decomposition.

6. PLOS authors have the option to publish the peer review history of their article (what does this mean?). If published, this will include your full peer review and any attached files.

Reviewer #1: No

---

## [Author Response · Author response to Decision Letter 0]

14 Dec 2022

Please see our response in the rebuttal letter.

---

## [Editor Report · Decision Letter 1]

19 Dec 2022

GAME BIRD CARCASSES ARE LESS PERSISTENT THAN RAPTOR CARCASSES, BUT CAN PREDICT RAPTOR PERSISTENCE DYNAMICS

PONE-D-22-26139R1

Dear Dr. Eric Hallingstad,

We’re pleased to inform you that your manuscript has been judged scientifically suitable for publication and will be formally accepted for publication once it meets all outstanding technical requirements.

Kind regards,

Dárius Pukenis Tubelis, Ph.D.

Academic Editor

PLOS ONE

Additional Editor Comments (optional):

Dear Dr Eric Hallingstad,

Congratulations on your excelent study.

I have read your corrected version, and I consider that it should be accepted

for publication in PLOS ONE. It is very well organised and written.

This is because you provided appropriate responses and changes to the reviewer´s suggestions.

The text was shortened as we requested, and it is easier to read now.

All figures and tables were appropriately cited and placed in the text.

The same occurs with the supplementary material, that reduced the text.

I found no typing errors in the text. Please check this again when correcting the proofs.

However, I found some minor problems in the references section.

Please point out these problems when you do the proof checking:

Ref. 03. Please add the DOI.

Refs 05 and 36. Delete one dot point after "et al". There are two, and should be only one.

Ref. 10. Delete the dot after "S".

Ref. 16. Delete the dot after "G".

Ref. 34. What year is correct ?

Refs. 48-50. They have DOIs. Please add. Just copy their titles to the Google to get them.

Ref. 54. Please delete "Accessed".

Add other necessary changes if you find them.

Dárius

---

## [Editor Report · Acceptance letter]

23 Dec 2022

PONE-D-22-26139R1 

Game bird carcasses are less persistent than raptor carcasses, but can predict raptor persistence dynamics 

Dear Dr. Hallingstad:

I'm pleased to inform you that your manuscript has been deemed suitable for publication in PLOS ONE. Congratulations! Your manuscript is now with our production department. 

Kind regards, 

on behalf of

Dr. Dárius Pukenis Tubelis 

Academic Editor

PLOS ONE